# The Role of Mitochondrial Dynamics and Mitotic Fission in Regulating the Cell Cycle in Cancer and Pulmonary Arterial Hypertension: *Implications for Dynamin-Related Protein 1 and Mitofusin2 in Hyperproliferative Diseases*

**DOI:** 10.3390/cells12141897

**Published:** 2023-07-20

**Authors:** Pierce Colpman, Asish Dasgupta, Stephen L. Archer

**Affiliations:** Department of Medicine, Queen’s University, Kingston, ON K7L 3N6, Canada; 16pcc1@queensu.ca

**Keywords:** pulmonary arterial hypertension, apoptosis, mitophagy, cancer, mitotic fission, mitochondrial dynamics protein of 49 kDa (MiD49), mitochondrial dynamics protein of 51 kDa (MiD51), mitochondrial fission protein 1 (Fis1), mitochondrial fission factor (MFF), dynamin-related protein 1 (Drp1), cardiolipin (CL), phosphatidic acid (PA), dynamin 2 (DNM2)

## Abstract

Mitochondria, which generate ATP through aerobic respiration, also have important noncanonical functions. Mitochondria are dynamic organelles, that engage in fission (division), fusion (joining) and translocation. They also regulate intracellular calcium homeostasis, serve as oxygen-sensors, regulate inflammation, participate in cellular and organellar quality control and regulate the cell cycle. Mitochondrial fission is mediated by the large GTPase, dynamin-related protein 1 (Drp1) which, when activated, translocates to the outer mitochondrial membrane (OMM) where it interacts with binding proteins (Fis1, MFF, MiD49 and MiD51). At a site demarcated by the endoplasmic reticulum, fission proteins create a macromolecular ring that divides the organelle. The functional consequence of fission is contextual. Physiological fission in healthy, nonproliferating cells mediates organellar quality control, eliminating dysfunctional portions of the mitochondria via mitophagy. Pathological fission in somatic cells generates reactive oxygen species and triggers cell death. In dividing cells, Drp1-mediated mitotic fission is critical to cell cycle progression, ensuring that daughter cells receive equitable distribution of mitochondria. Mitochondrial fusion is regulated by the large GTPases mitofusin-1 (Mfn1) and mitofusin-2 (Mfn2), which fuse the OMM, and optic atrophy 1 (OPA-1), which fuses the inner mitochondrial membrane. Mitochondrial fusion mediates complementation, an important mitochondrial quality control mechanism. Fusion also favors oxidative metabolism, intracellular calcium homeostasis and inhibits cell proliferation. Mitochondrial lipids, cardiolipin and phosphatidic acid, also regulate fission and fusion, respectively. Here we review the role of mitochondrial dynamics in health and disease and discuss emerging concepts in the field, such as the role of central versus peripheral fission and the potential role of dynamin 2 (DNM2) as a fission mediator. In hyperproliferative diseases, such as pulmonary arterial hypertension and cancer, Drp1 and its binding partners are upregulated and activated, positing mitochondrial fission as an emerging therapeutic target.

## 1. Introduction

Mitochondria are semi-autonomous organelles which have an ancestral relationship to alpha-proteobacterium [1]. Mitochondria contain their own genome and protein synthesis machinery and have been coopted by eukaryotic cells in a symbiotic relationship that achieves many crucial cellular functions [2]. Mitochondria are best known for their ability to generate energy in the form of ATP through the process of oxidative phosphorylation and are actively involved in many other catabolic and anabolic reactions, including the citric acid cycle, beta-oxidation of fatty acids, maintenance of calcium homeostasis, and the biosynthesis of haem, phospholipids and other metabolites [3]. In addition, mitochondria are key regulators of apoptosis, cell-cycle progression, cellular proliferation, differentiation of cells and mitophagy [4].

Mitochondria are dynamic organelles which translocate within the cell, moving along tubulin tracks. Furthermore, they continuously divide and fuse through highly regulated processes called fission and fusion [5]. Mitochondrial fission, fusion and translocation are collectively known as mitochondrial dynamics (Figure 1). Mitochondrial dynamics are controlled by nuclear encoded enzymes, mostly large GTPases [4], as well as mitochondrial lipids, such as cardiolipin and phosphatidic acid [6].

Because mitochondria generate substantial reactive oxygen species (ROS) and have limited DNA repair mechanisms, acquired mutations of mitochondrial DNA (mtDNA), such as ROS-induced damage of mitochondrial proteins and lipids, are common [7]. Mitochondrial fusion is a compensatory mechanism by which mitochondria reduce oxidative stress and lessen the resultant mutation burden [8]. This process, termed complementation, reflects the sharing of mitochondrial DNA between healthy and damaged mitochondria. When a mitochondrion with largely normal mtDNA fuses with a mitochondrion that has mutated or damaged mtDNA, the healthy organelle donates sufficient normal mtDNA to overcome the respiratory defects of the damaged organelle, creating a single healthy mitochondrion. Nakada et al., showed that fusion of respiration-deficient and healthy mitochondria enabled the exchange of genetic contents and restored mtDNA-based polypeptide synthesis due to the actions of the normal tRNA transcribed from the healthy, ‘parental’, mitochondria [9]. When healthy and diseased mitochondria are crossed with each other, daughter cells may show recovery of the wild-type phenotype, a manifestation of “genetic complementation” [10]. Complementation is effective provided the mutation load remains below ~80% per cell [11]. Maternally inherited, double stranded, mtDNA is packaged in nucleoids, discrete complexes surrounded by proteins [12]. Nucleoids are unable to transfer mtDNA between themselves [13], therefore it is likely that mitochondria complement each other by sharing mtRNA and proteins [7].

Mitochondrial fission has many functional implications. Fission can create new mitochondria (mitochondrial biogenesis) and also allows the distribution of mitochondria to daughter cells during mitosis (mitotic fission). In addition, fission enables the removal of damaged portions of a mitochondrion from the network through mitophagy. During high levels of cellular stress, fission can promote apoptosis which eliminates damaged cells, another form of cellular quality control [7]. Acquired abnormalities of mitochondrial dynamics are increasingly recognized as contributing to the pathogenesis of human diseases [14,15], ranging from neurodegenerative conditions, such as Alzheimer’s disease [16], to cancer [17,18,19], and cardiopulmonary disease, such as pulmonary arterial hypertension (PAH) [20,21] and cardiac ischemia reperfusion injury [22,23].

Mitochondrial dynamics are largely controlled by large guanosine triphosphatases (GTPases) in the dynamin family [24]. Mitochondrial fusion is mediated by mitofusins 1 and 2 (Mfn1/2) and optic atrophy-1 (Opa1). Mitochondrial fission is mediated by dynamin-related protein 1 (Drp1). Unlike fusion mediators such as Mfn1 and Mfn2, which exist in the outer mitochondrial membrane (OMM), and OPA1, which exists on the inner mitochondrial membrane (IMM), Drp1 is predominantly found in the cytosol and moves to mitochondria once it is activated by post translational Drp1 modification, notably changing in phosphorylation at sites such as serine 616 [25]. Drp1 moves dynamically between the cytoplasm and mitochondria with ~53% of Drp1 in the cytoplasm and the rest residing on mitochondria [26]. Once activated, Drp1 translocates to the mitochondria where it binds to one or more adapter proteins on the OMM (e.g., Fis1, MFF, MiD51 or MiD49). These Drp1 binding partners focus the recruitment of Drp1 to specific sites on the OMM and regulate fission. In addition to Drp1 and its binding partners, the endoplasmic reticulum (ER), actin cytoskeleton and membrane phospholipids in the mitochondria participate in mitochondrial fission. Recently a role has been suggested for dynamin 2 (DNM2), another large GTPase, as a mediator of the terminal stage of fission [27], although this remains controversial [28].

## 2. Mitochondrial Fusion

In mammals, Mfn1 and Mfn2 (Table 1) are located on the OMM and each isoform contributes to fusion [29]. Both Mfns contain an N-terminal GTP binding domain, followed by a heptad repeat coiled-coil region (HR1), two transmembrane segments and a second coiled-coil domain which has a heptad repeat (HR2) at the C-terminus (Figure 2) [30]. Key functional domains are conserved between Mfn1 and Mfn2, with the GTPase and two heptad repeats facing the cytosol in both. To fuse adjacent mitochondria, mitofusins hydrolyze guanosine triphosphate (GTP) using the N-terminal GTPase domain (Figure 2) [31]. The trans model of mitochondrial fusion involves the HR2 domain nearest the C-terminus interacting with the HR2 domain on an adjacent mitochondrion that is destined for fusion (Figure 2). This dimeric, antiparallel interaction between the two mitochondria’s HR2 domains, achieves docking, followed by GTP hydrolysis and ultimately OMM fusion [32,33].

Mfn1 and Mfn2 can form both homotypic and heterotypic complexes in mouse embryonic fibroblast (MEF) cells [29], each of which can execute OMM fusion [29,36]. Each of the three types of Mfn complex participates in fusion, and the disruption of either Mfn1 or Mfn2 leads to mitochondrial fragmentation [29]. In Mfn2 ^−^/^−^ and Mfn1 ^−^/^−^ cells, overexpression of either Mfn1 or Mfn2 homotypic oligomers rescued mitochondrial fusion, suggesting redundancy in the role of mitofusin isoforms in these cells [29]. However, there is also cellular heterogeneity in the importance of the Mfn isoforms. For example, in trophoblast giant cells, only mitochondria from Mfn2-deficient embryos are fragmented, whilst mitochondria morphology was unaltered in cells from Mfn1-deficient embryos [29].

There are other important functional differences between Mfn1 and Mfn2. For example, only Mfn2 is expressed on the surface of the ER at mitochondria–ER contact sites, called mitochondrial-associated membranes (MAM’s). These sites denote the location of impending fission and are sites for calcium signaling between the ER and mitochondrion [37]. Mfn2 also has an N-terminus Ras-binding domain which is lacking in Mfn1 (Figure 2). Such structural differences suggest that Mfn2 may have roles in addition to fusion, and these roles may differ from those served by Mfn1 [38]. For example, Mfn2, but not Mfn1, is involved in energy metabolism, insulin signaling, mitophagy, apoptosis and in mitochondrial–ER fusion contact sites [39]. Mfn2 also participates in mitochondrial oxidation through mechanisms beyond those which promote mitochondrial fusion [40]. Mfn2 gain and loss of function changes the expression of nuclear encoded electron transport chain (ETC) subunits. Loss of Mfn2 function inhibits the expression of p39 (in complex I), p70 (in complex II), p49 (in complex III) and the alpha subunit of complex V. Furthermore, loss of Mfn2 reduces oxygen consumption, depolarizes mitochondrial membrane potential, and inhibits oxidation of glucose, pyruvate and fatty acids [40]. Conversely, gain of Mfn2 function induces the expression of p39, COX-I and COX V in ETC IV and the alpha subunit of complex V [41]. Through its actions on these ETC subunits, Mfn2 gain-of-function increases glucose oxidation and hyperpolarizes mitochondrial membrane potential [40].

IMM fusion follows OMM fusion [42] and is mediated by OPA1 [43,44]. OPA1 is attached to the IMM via an N-terminus transmembrane domain, which is downstream from its mitochondrial matrix targeting sequence, and it faces the IMM space. OPA 1 requires Mfn1, but not Mfn2, to accomplish mitochondrial fusion [45]. OPA1 silencing causes fragmentation of mitochondria whilst heterologous OPA1 overexpression causes mitochondrial elongation [46]. As a result of alternative splicing, at least eight variants of OPA1 exist, including several long (L-OPA1) and short (S-OPA1) isoforms [47]. A combination of L-OPA1 and S-OPA1 is essential for efficient membrane fusion in mammals; however, the mechanism by which cooperation occurs is not well understood. S-OPA1 can assemble in dimers on the IMM, creating a helical array with a dynamin-like structure that is critical in IMM fusion. Binding of GTP to S-OPA1 triggers a conformational change resulting in mitochondrial tubulation [48] such as occurs with GTP binding to S-Mgm1, the yeast homolog of OPA1. This suggests that S-OPA1 uses a GTP-dependent, dynamin-like, power stroke to accomplish IMM fusion [49]. Paradoxically, an excess of S-OPA1 inhibits fusion [50]. Heterologous overexpression of OPA1 counteracts the effects of Mfn2 silencing (but cannot compensate for the loss of Mfn1), a reminder of the functional differences between the two mitofusins [45].

Fragmentation of the mitochondrial network can occur due to decreased expression or function of OPA1, Mfn1 or Mfn2 [37]. Ishihara et al. reported that loss of Mfn1 in HeLa cells causes more mitochondrial fragmentation than the loss of Mfn2. They attributed this to Mfn1’s ~eight-fold higher GTP-hydrolysis-dependent, membrane-tethering, activity [29,51]. Conversely, Mfn2 has a higher affinity for GTP than does Mfn1 [51]. Homotypic Mfn1 interactions with mitochondria are 100-fold more frequent than those which occur with Mfn2 [51]. The importance of Mfn1 versus Mfn2 likely varies between cell type and amongst diseases. For example, in PAH pulmonary artery smooth muscle cells (PASMC), Mfn2, but not Mfn1, is downregulated, and augmenting Mfn2 restored fusion and regressed PAH in a preclinical female rat model of PAH [52].

## 3. Mitochondrial Fission

Drp1 and its binding partners constitute the key mitochondrial fission machinery in mammals, but there are other established and putative fission mediators. For example, the ER and actin pre-constrict mitochondria at sites of impending fission, and DNM2 may mediate the final step that divides the mitochondria (although this remains controversial). A role for other fission mediator mechanisms in addition to Drp1 is suggested by basic considerations of mitochondrial size. Mitochondria are too large for Drp1 multimers to initiate constriction, and the minimum diameter that can be achieved by the homomultimeric Drp1 fission apparatus is insufficient to fully divide the organelle. In human cells, mitochondrial tubes have a diameter of ~300 nm [53]. Long et al. showed the mean diameter of mouse embryonic fibroblast (MEF) mitochondria is 260 ± 70 nm, as measured by TEM, whilst STORM and PALM optical microscopy measured mitochondrial diameters of 290 ± 50 nm and 210 ± 30 nm, respectively. Thus, mitochondrial diameter ranges from ~200–300 nm [54]. This physical scale is problematic in understanding how Drp1 initiates fission since the assembled Drp1 macromolecular scission ring is predicted to be smaller than the diameter of typical mitochondria. Drp1 ring diameter is initially 139 ± 60 nm prior to constriction and 48 ± 38 nm post fission [55]. This size mismatch is addressed by a constriction step prior to Drp1 assembly, which reduces the mitochondrial diameter to ~150 nm to accommodate Drp1 [56]. Friedman et al. found that the ER accomplishes this pre-constriction and defines the site of a future fission event [56].

Drp1, once activated by post translational modification, translocates to the OMM where it interacts with various receptor proteins on the OMM (Fis1, MFF, MIEF1/MiD51 and MIEF2/MiD49) (Figure 3) [4]. Under physiological conditions Drp1 coexists in a dynamic equilibrium between a cytosolic form and a membrane-bound form, which equilibrate through a dimeric intermediate [57]. Drp1 dimers, the smallest unit of assembly that retains Drp1 activity, potentiate the assembly and organization of membrane-bound Drp1, which is required for membrane fission [57]. At the OMM, Drp1 is assembled into a higher-order, multimeric complex. This ring-like oligomer wraps around the mitochondrion, triggering fission through its GTPase activity (Figure 3) [57,58]. The GTPase domain of Drp1 is essential for its fissogenic function. Consistent with this, overexpression of a mutant Drp1 that has a dysfunctional GTPase domain in a Drp1 silenced cell, does not restore mitochondrial fission (Figure 4) [23].

Drp1 does not have a membrane anchoring domain, and to be recruited to the OMM for fission, Drp1 must be activated. Regulation of Drp1 activity is mediated by site-specific phosphorylation and dephosphorylation by various kinases and phosphatases. Substantial evidence shows that Ser616 phosphorylation promotes Drp1 activation and translocation from the cytosol to the OMM, while phosphorylation at Ser637 reverses this process (reviewed in [59]. Phosphorylation and activation of Drp1 is accomplished by numerous kinases, including phosphoglycerate mutase family member 5 (PGAM5) [60], AMP-activated protein kinase (AMPK) [61,62], mitogen-activated protein kinase (MAPK) [63], cyclin-dependent kinases/cyclin B1 (CDK/Cyclin B1), and Aurora kinase A [64]. In contrast, the phosphatase calcineurin promotes fission through the calcium-dependent dephosphorylation of Drp1 (Ser637) [65]. One mechanism by which phosphorylation of serine 637 inhibits fission is by preventing binding with MiD49 [66]. These post translational regulatory mechanisms link fission to numerous cellular functions, including the regulation of the cell cycle.

There is a second “size paradox” that pertains to mitochondrial fission. Drp1’s oligomerized ring-like structure has a minimum inner diameter of 15 nm [57,67], and it is debated whether Drp1 constriction is sufficient on its own to accomplish both outer and inner membrane fission. It has been proposed that the final scission event in mitochondrial fission is carried out by DNM2 (Figure 5), an endocytic-related GTPase [53,68]. Lee et al. found that DNM2 was recruited to ER-induced mitochondrial fission sites and showed that silencing of DNM2 in human HeLa and monkey Cos-7 cells, using small interfering RNA (siRNA), caused mitochondrial hyperfusion and apoptosis resistance [27]. This observation has not yet been corroborated. In fact, other groups have recently reported opposite findings, that silencing dynamin expression does not affect mitochondrial fission. Kamerkar et al. and Fonseca et al. report that mouse fibroblasts lacking all three mammalian dynamin proteins (DNM1, DNM2 and DNM3), as well as cells with knockdown of DNM2 only, do not display defects in mitochondrial or peroxisomal fission. They concluded that Drp1 can achieve mitochondrial and peroxisomal fission in the absence of dynamins 1–3 and concluded that dynamins are not an essential part of the mitochondrial fission process [28,69]. Therefore, the ultimate role of DNM2 in mitochondrial fission remains controversial.

A pathway which involves a contact site between trans Golgi network (TGN) vesicles, mitochondria and phosphatidylinositol 4-phosphate (PI4P) signaling has also been proposed to contribute to the final steps of mitochondrial fission [72]. ADP-ribosylation factor-1 (Arf1) is a member of the Arf small GTPase family within the Ras superfamily. Arf1 mediates membrane remodeling and vesicle formation by controlling the recruitment of protein effectors to these sites [73]. At the TGN, Arf1 recruits PI4P kinase which phosphorylates PI, forming PI4P and facilitating vesicle formation and release [74]. Microdomains of PI4P on TGN vesicles localize to areas of mitochondria–ER contact and can drive mitochondrial division downstream of Drp1 [72]. The loss of Arf1 or its effector, phosphatidylinositol 4-kinase IIIβ (PI4KIIIβ) reduces PI4P generation and results in hyperfused, branched mitochondria with extended mitochondrial constriction sites [72]. Thus, recruitment of vesicles containing TGN-PI4P to mitochondria–ER contact sites may facilitate the final events in mitochondrial fission [72].

The impact of mitochondrial fission is contextual. In the mitochondrial life cycle, fission variably enables the biogenesis of new mitochondria, cell cycle progression, and the clearance of dysfunctional mitochondria through mitophagy or elimination of abnormal cells through apoptosis. Historical models of mitochondrial fission cannot explain the mechanism for these diverse consequences of fission. However, Kleele et al. recently posited that the relative positioning of the fission site along the length of the mitochondrion is a key determinant of whether fission is involved in mitochondrial proliferation or degradation (Figure 6) [70]. Live-cell structured illumination microscopy identified two functionally and mechanistically distinct types of fission. Functionally, the division of mitochondria at its periphery was associated with quality control and led to the removal of damaged portions of the mitochondria by mitophagy. In contrast, mitochondrial midzone fission was associated with the proliferation of mitochondria, as would occur in mitotic fission (Figure 6). Mechanistically, both types of fission required Drp1; however, midzone fission was aided by MFF and actin and occurred at a site of contact with the ER, whereas peripheral fission was mediated by Fis1 and occurred at a site of lysosomal contact [70].

Upstream of Drp1, actin polymerization coordinates with mitochondrial–ER contacts to define sites of fission and trigger mtDNA replication [56,75]. However, not all fission is associated with ER contact. While midzone fission sites are consistently contacted by the ER before fission, most peripheral fission sites were not (Figure 6) [70]. Additionally, immunostaining for PDZD8, a mitochondria–ER tethering protein, revealed more PDZD8 at midzone versus peripheral fission sites (Figure 6) [70]. The conditional involvement of ER contacts in midzone, but not peripheral, fission events support previous reports that not all mitochondrial fission events involve ER contacts, nor are they always associated with mtDNA replication [76,77]. If confirmed, this would suggest that different composition and location of Drp1-containing fission complexes may account for the observed contextual differences in the impact of fission [70].

## 4. Mitochondrial Receptors Involved in Drp1 Recruitment to the OMM

The recruitment of Drp1 to the mitochondria and the organization of Drp1 on the OMM is primarily mediated by four Drp1 receptor proteins anchored on the OMM (Fis1, MFF, MIEF1, also known as MiD51, and MIEF2, also known as MiD49).

### 4.1. Fis1

Mitochondrial fission 1 (Fis1) is attached to the OMM through its C-terminal transmembrane domain, which has a cytosolic-facing N-terminal region that contains an N-terminal tetratricopeptide repeat (TPR) motif [78,79]. Although Fis1’s TPR is atypical, it is nonetheless thought to be important for protein binding [80]. Fis1 overexpression promotes fission and can induce mitochondrial fragmentation in mammalian cells; conversely Fis1 inactivation promotes a hyperfused mitochondrial morphology in some cell types [79,81]. However, Fis1’s role in fission is contextual, varying by disease state and cell type. For example, while Fis1 is involved in mediating pathologic mitochondrial fission and ROS production in cardiac ischemia reperfusion (IR) injury [82], Fis1 is not critical to the increased mitotic fission seen in proliferating PAH PASMCs [20]. Tian et al. studied the effects of mdivi-1 (a Drp1 inhibitor) and P110 (a competitive peptide inhibitor of the Drp1–Fis1 interaction) on mitochondrial morphology, mitochondrial membrane potential and RV function before and after two cycles of ischemia reperfusion injury challenge (RV-IR) (Figure 7). In RV-IR, P110 was as effective as Mdivi-1 in preserving mitochondrial structure and improving RV diastolic function (Figure 7). These findings indicate that Fis1 contributes to the pathologic fission in RV-IR [82].

Interestingly, Fis1 can interact with Mfn2 and OPA1 to mediate mitochondrial fission in the absence of Drp1 by impairing their GTPase activity and inhibiting fusion, leading to mitochondrial fragmentation [83]. Fis1’s role in fission is debated and varies between cell types. In HeLa cells, Fis1 depletion elongates mitochondria promoting a fused network [84] but also has been found to have no effect on mitochondrial fragmentation by others [85]. Fis1 depletion also had no effect on mitochondrial morphology in a human colorectal carcinoma cell line, HCT116 [86], consistent with our finding that MiD49 and MiD51 are more important than Fis1 in the mechanism of mitochondrial fission in cancer cells [87]. Additionally, speaking to the fact that Fis1 is not critical to fission in many cells, the concentration of Fis1 in HeLa cells does not influence the distribution of Drp1 between the cytosol and OMM [80,85]. Interestingly, although elevated levels of Fis1 in HeLa cells did not affect the subcellular distribution of Drp1, mitochondrial fission did increase [88]. Furthermore, knockdown of Fis1 in HeLa and HCT116 cells did not reduce Drp1 levels on the OMM [85]. In contrast, Fis1 ^−^/^−^ MEFs did have reduced mitochondrial Drp1 levels on the mitochondrial OMM [89]. Thus, there is cellular heterogeneity in the contribution of Fis1 to fission and to Drp1 recruitment to the OMM.

### 4.2. MFF

Mitochondrial fission factor (MFF) also contributes to the recruitment of Drp1 to the OMM, although, as with Fis1, there is heterogeneity in the contribution of MFF between cell types [86]. The *MFF* gene generates nine different isoforms through alternative splicing [90]. MFF’s C-terminal transmembrane domain anchors it to the OMM much like Fis1. MFF’s N-terminal region faces the cytosol and contains three small amino acid repeats (R1-R3) and a coiled-coil domain [90]. The R1 and R2 motifs, located within the initial 50 N-terminal amino acid residues, are critical to the recruitment of Drp1 to the OMM, and without this region Drp1-MFF interaction cannot occur [86]. In HeLa cells or MEFs, depletion of MFF drastically reduces mitochondrial fission and impedes Drp1 recruitment to the OMM. Conversely, MFF overexpression in these cells causes Drp1 recruitment to the mitochondria and induces hyper-fission [86]. MFF-mediated mitochondrial fission occurs independently of Fis1, further strengthening the notion that Fis1 is dispensable for Drp1 recruitment in some cells [86]. The relative importance of MFF versus Fis1 as a receptor for Drp1 recruitment requires further study, with careful attention to cell type and disease state as determinants of the precise molecular mediators of fission. The capacity of MFF to promote fission is regulated at a post translational level. For example, 5’AMP activated protein kinase (AMPK), which is activated by mitochondrial stress, triggers mitochondrial fission through the phosphorylation of MFF at serine residues 155/172. AMPK-induced phosphorylation of MFF enhances the recruitment of Drp1 to the OMM and increases mitochondrial fission. AMPK-enhanced, MFF-mediated fission underlies fragmentation of human U2OS osteosarcoma mitochondria in vivo [91]. Consistent with this, silencing of MFF in HeLa cells inhibits mitochondrial fission [90]. Silencing MFF also inhibited staurosporine-induced cytochrome c release from mitochondria and reduced apoptosis in HeLa cells [90]. MFF is also involved in peroxisomal division and siMFF inhibits peroxisomal fission in HeLa cells [90].

### 4.3. MiD51 and MiD49

MiD51 and MiD49 are paralogs that are mitochondrial receptors for cytosolic Drp1 translocation to mitochondria in mammals [88,92]. Also known as MIEF1 and MIEF2, respectively, these receptors are highly conserved in vertebrates but have yet to be identified in invertebrates and plants [93]. MiD51 and MiD49 have a 45% amino acid homology in humans [4]. MiD proteins have N-terminal transmembrane domains which anchor them to the OMM where they form homo- or heterodimers [93]. MiD51 exists primarily as a dimer, whereas MiD49 occurs in both monomeric and oligomeric forms [4]. While both MiDs have nucleotidyl transferase domains, only MiD51’s domain can bind to nucleotide diphosphates, such as ADP and GDP [94,95]. When MiD51 binds ADP, it stimulates Drp1 oligomerization and assembly and activates Drp1’s intrinsic GTPase activity [96].

Overexpression of MiD51 or MiD49 leads to recruitment of cytosolic Drp1 to the OMM through a Fis1-independent mechanism [97]. MiD proteins are more efficient at recruiting Drp1 than is MFF or Fis1 [98]. However, conflicting results exist as to the consequences of overexpression of MiD51/49 on mitochondrial fission. Losón et al. found that overexpression of MiD51/49 induces mitochondrial elongation, possibly by inhibiting fission [89], whereas overexpression of MFF increased mitochondrial fragmentation [86,90]. They interpreted this as indicating that MFF is the principal receptor of Drp1 in mitochondrial fission [86,99]. In contrast, it is reported that overexpression of MiD51 acts as a Drp1 inhibitor by reducing Drp1’s binding to GTP [88,98]. However, we consistently demonstrated that increased MiD expression, whether occurring spontaneously, in the course of a human disease or by heterologous overexpression, increases mitochondrial fission. MiD expression is increased in PASMC from patients with PAH, and this upregulation of MiD49 and MiD51 increases fission and promotes a hyperproliferative, apoptosis-resistant, cancer-like PAH phenotype (Figure 8) [20]. MiD49 and MiD51 are also upregulated in the pulmonary arteries and lungs of rats with experimental PAH. Likewise, heterologous expression of MiDs in normal PASMC was sufficient to create a PAH phenotype with increased Drp1-dependent mitochondrial fission and accelerated cell proliferation (Figure 8). Conversely, silencing MiDs in PAH PASMC increased mitochondrial fusion, slowed proliferation and induced apoptosis [20]. Consistent with this, we showed that silencing MiDs in vivo using siRNA, regressed established PAH in a preclinical animal model [20]. Upregulation of MiDs in PAH (human and experimental) is caused by an epigenetic mechanism, namely a pathological decrease in microRNA miR-34a-3p levels. Plasma miR-34a-3p levels are low in PAH patients, suggesting this miR has potential value as a PAH biomarker. An SNP (exm1300952) in the gene encoding MiD49 [100] has been identified in patients with WHO Group 2 PH, suggesting MiD49 may have relevance to other forms of pulmonary hypertension.

MiDs serve as mediators of fission and regulators of the cell cycle and apoptosis in cancer [87]. MiDs have a role in the pathogenesis of invasive breast cancers (IBC) and non-small cell lung cancers (NSCLC). Increased MiD expression is a driver of increased fission, cell proliferation and apoptosis-resistance in NSCLC and IBC [87]. MiD expression is pathologically upregulated in cancer cells (NSCLC and IBC) by the same epigenetic mechanism as occurs in PAH decreased miR-34a-3p expression. Silencing MiDs inhibited fission and caused cell cycle blockade in human cancer cells by inhibiting the Akt–mTOR–p70S6K pathway. Thus, both MiD49 and MiD51 are critical participants in mitotic fission in cancer. In vivo, silencing MiDs regressed NSCLC tumors in a xenotransplant NSCLC model. In patients with breast cancer, a strong positive correlation was observed between increased MiD expression and larger tumor size and more severe tumor grade. Conversely, an inverse correlation was observed between MiD expression and survival in patients with NSCLC. Thus, the microRNA-34a-3p-MiD axis is critical to cancer pathogenesis and constitutes a new therapeutic target and potential biomarker [87]. These studies indicate MiDs to be “pro-fission” in normal and diseased human cells. These findings are in opposition to the findings of Losón et al. [89], although it is uncertain why.

## 5. Communication between Mitochondrial Fission and Fusion Mediators

Mitochondrial morphology reflects the balance between fission and fusion. While it was initially thought that fission and fusion were discrete events, recent evidence suggests interplay between these two mechanisms. For example, fragmented mitochondria are rare in Drp1 overexpressing cells [101,102]. Furthermore, most mitochondria in Drp1 overexpressing MEF cells are elongated and collapsed around the nucleus suggesting that the ability of Drp1 overexpression to cause fission is limited by the number of Fis1/MFF molecules. When adequate Fis1 and MFF are not available, Drp1 may paradoxically increase fusion through interactions with Mfn2 [103]. Drp1 colocalizes with Mfn2 and Bax at mitochondrial fission sites during apoptosis [104], providing additional evidence that Drp1 interacts with Mfn2. In this study, Mfn1 and Mfn2 single knockout MEF cells were transfected with wild-type Drp1, and mitochondrial morphology was assessed. Drp1 overexpression paradoxically restored an elongated tubular mitochondrial phenotype, whereas mitochondria in adjacent, untransfected cells were highly fragmented. Additionally, in Mfn2 KO cells Drp1 overexpression resulted in elongated mitochondria indicating that Drp1 can interact with Mfn1. The heightened effect of mitochondrial elongation seen in cells containing only Mfn1 suggests that Drp1 may preferentially interact with Mfn1 or that Mfn1 may be more available or expressed in a higher concentration in the fibroblast cell line that was studied. Drp1 is near Mfn2 at the mitochondrial surface, indicating the proximity of both fission and fusion microenvironments at shared locations [104]. Thus, Drp1 may enhance mitochondrial fusion through interactions with mitofusins, in addition to its canonical pro-fission role [103]. There is precedent for noncanonical interactions between regulators of mitochondrial fission and fusion. For example, Fis1 can interact with Mfn2 in Drp1-deficient conditions promoting fission [83], whilst S-OPA1 can localize with the fission machinery and promote fission [105].

## 6. Membrane Phospholipids and Mitochondrial Dynamics

The phospholipids, phosphatidic acid (which promotes fusion) and cardiolipin (which promotes fission) modulate mitochondrial dynamics [106]. In mammalian cells, phosphatidic acid accounts for just 1–2% of total lipid and cardiolipin 2–5%; however, the lipids are highly enriched in mitochondria [107]. Cardiolipin is exclusively generated in the IMM by cardiolipin synthase from phosphatidic acid. Cardiolipin can be transported to the OMM where it interacts with Drp1 through its variable domain (also called the B-insert) (Figure 9) [108]. This interaction stimulates Drp1 oligomerization and GTPase activity, thereby enhancing fission [6,109]. Interestingly cardiolipin-increased Drp1 GTPase activity is synergistically enhanced in the presence of MFF, suggesting that cardiolipin interacts with MFF to enhance mitochondrial fission [106].

Once in the OMM, cardiolipin can be transformed back to phosphatidic acid by mitochondrial phospholipase D (MitoPLD) [107]. The resulting PA-rich environment near the division apparatus inhibits Drp1 activity and leads to mitochondrial fusion (Figure 9) [110]. Drp1 recognizes the head group of phosphatidic acid and two saturated acyl chains of other, adjacent, non-PA phospholipids by penetrating the hydrophobic core of the membrane [6]. These dual phospholipid interactions inhibit Drp1 GTP hydrolysis, preventing fission [6]. Phosphatidic acid is transported from the ER to the IMM and ultimately participates in cardiolipin synthesis.

Dudek notes that cardiolipin is preferentially found in regions of locally high membrane curvature, such as cristae, where it preferentially resides in the monolayer facing the mitochondrial matrix [111]. Phosphatidic acid affects the biophysical properties of bilayer membranes and facilitates negative membrane curvature through its small, negatively charged, head group [112]. Negative curvature makes biological membranes prone to fusion, and thus phosphatidic acid is a fusogenic phospholipid [113]. MitoPLD overexpression induces fusion, while inhibition of Mito-PLD results in mitochondrial fragmentation. In addition, Lipin 1b, a phosphatase which converts phosphatidic acid to diacylglycerol, induces mitochondrial fragmentation [114]. Drp1’s intrinsically disordered variable domain can undergo a helical structural transition which allows it to penetrate the membrane bilayer and selectively bind cardiolipin [115]. When the Drp1 domain that binds cardiolipin is mutated, “donut”-shaped circular mitochondria are generated, which may be important in stress-induced fission [115].

## 7. The Endoplasmic Reticulum and Mitochondrial Dynamics

The ER uses mitochondria-associated membranes (MAMs) to communicate with mitochondria and support phospholipid synthesis, protein import, mitochondrial calcium uptake from the ER, and mitochondrial fission [116]. Mfn2 is found both on the mitochondria and the ER and connects these organelles, maintaining sufficient mitochondrial calcium uptake to support oxidative ATP synthesis [116]. Electron microscopy shows that mitochondria are surrounded by ER tubules, which lie within 200 nm of the OMM [117]. In addition, digital, 3D deconvolution microscopy studies indicate that as much as 20% of the entire surface of the mitochondrion is in contact with the ER [118]. The ER pre-constricts the mitochondria prior to Drp1 recruitment, identifying the location of impending fission and “right-sizing” the mitochondria to allow Drp1 multimers to encircle the mitochondria [56]. ER tubules principally position themselves at Drp1 and MFF foci on the mitochondrion but also denote positions of milder constriction when MFF or Drp1 are absent [56].

## 8. The Actin Cytoskeleton, Actin Motors and Mitochondrial Dynamics

The assembly and disassembly of actin near mitochondria promotes Drp1-dependent mitochondrial fission [119,120]. Inverted formin 2 (INF2) is an ER-localized actin regulator that causes actin filaments to assist in the initial mitochondrial constriction which subsequently promotes Drp1 recruitment to the ER-mitochondria constriction site [75]. The actin-nucleating protein, Spire1C, is localized to the OMM and interacts with INF2 to promote actin assembly. INF2 and Spire1C use ER–mitochondrial membrane contact points, respectively, as a scaffold for assembly of actin cables. These actin cables are thought to provide the mechanical force required to induce initial membrane constriction and mark the sites for subsequent Drp1 recruitment [75,121] (Figure 5). Defects or dysfunction in Spire1C, that reduce its formin binding activity or actin nucleation, impair mitochondrial constriction and prevent fission [121].

The distribution of mitochondria and their motility is regulated in part by intracellular calcium fluctuations, [Ca2+]c [122]. Calcium-dependent regulation of the Miro–Milton complex controls both mitochondrial motility and the balance between fission and fusion by linking mitochondria with kinesin motors [123]. Mammals have two Miro family members, Miro1 and Miro2, which share 60% sequence homology. These Miros have N- and C-terminus GTPase domains separated by two calcium binding EF hand motifs, as well as a C-terminus transmembrane domain, which aids in targeting the protein to the OMM [124]. The calcium sensitivity of Miros is due to their calcium binding EF-hand motifs [125]. Milton family proteins are cytoplasmic and aid in mitochondrial transport. Milton binds to Miro and to the kinesin heavy chain (KHC), thereby participating in mitochondrial motility [126]. Milton and Miro localize to the mitochondria in mammalian neurons [127]. Association of KHC’s motor domain with Miro occurs when [Ca2+]c is elevated [128]. However, Milton also connects KHC’s tail domain to Miro on mitochondria in a calcium-independent manner [126,129]. Consistent with this, silencing of Miros suppresses mitochondrial motility whilst Miro overexpression increases mitochondrial motility even when [Ca2+]c is maintained at a normal resting level (<100 nM) [130].

## 9. Mitochondrial Dynamics and Apoptosis

Mitochondrial dynamics are important in the regulation of cell death [131]. Extensive mitochondrial fragmentation is seen during apoptosis, and several mitochondrial fusion/fission proteins are implicated [132]. Current theories of how mitochondrial morphology and fission and fusion dynamics relate to apoptosis are controversial, with discordant findings in the literature. Some of the confusion relates to failure to consider the cell types studied (somatic cells which are prone to apoptosis versus cancer cells which are resistant to apoptosis) and/or failure to specify whether the observed effects relate to unstimulated apoptosis vs. chemically induced apoptosis. Mitochondrial fission can promote apoptosis in pathological conditions, such as cardiac IR injury [82]. However, in rapidly dividing PAH PASMC and cancer cells (breast and lung), fission is increased while apoptosis is suppressed. Inhibiting Drp1 in these cells promotes cell cycle arrest and increases apoptosis [20,87]. Similar apoptosis induction occurs in various lung cancer cell lines when inhibiting Drp1; using siDrp1, small molecule Drp1 inhibitors such as mdivi-1 [133] or drpitor1a [23]; or augmenting Mfn2 [133]. Thus, forcing a sustained state of fusion leads to apoptosis in dividing cells. Similarly, overexpression of Mfn2 can inhibit cell proliferation and induce apoptosis of rat vascular smooth muscle cells (VSMC) [134]. Debate remains as to whether the anti-proliferative effects of Mfn2 require mitochondrial fusion.

Conversely, there are reports indicating that hyperfused mitochondrial networks confer resistance to apoptosis [135]. Similarly, knockdown of Fis1 or Drp1 can lead to mitochondrial hyperfusion and resistance to apoptotic stimuli [132], whilst cells lacking fusion proteins (Mfn1, Mfn2, or OPA1) exhibit heightened mitochondrial fission and cellular sensitivity to apoptotic stimuli [136]. Additionally, when OPA1 is knocked down, mitochondrial fusion is halted and mitochondrial fragmentation occurs, along with depolarization of the mitochondrial membrane. Subsequent alterations in IMM structure result in cytochrome c release from mitochondria, inducing caspase activation and apoptosis [42,85].

The mitochondrial network likely assumes different morphologies at different stages in the apoptotic process (fused initially and fragmented as apoptosis nears its conclusion). Increased fission and decreased fusion causes mitochondrial fragmentation during apoptosis and this is associated with Bax translocation to mitochondria and cytochrome c release, followed by caspase activation [137]. In healthy cells, Bax, a pro-apoptotic protein of the Bcl-2 family, is localized largely in the cytosol. However, during apoptosis, Bax translocates to the surface of the mitochondria [138]. At the OMM, Bax coordinates with Bak, another pro-apoptotic Bcl-2 family protein, and increases OMM permeabilization [138]. Fis1 is required for this Bax translocation [85]. Upon apoptotic stimulation, Drp1 is recruited to the OMM where it co-localizes with Bax and Mfn2 at fission sites [104]. Drp1 promotes the activation and oligomerization of Bax [139]. This function of Drp1 is independent of its GTPase activity and relies instead on the interaction between Drp1’s arginine 247 residue and cardiolipin in membranes [140], which serves as a “binding platform” to recruit apoptotic factors such as tBid, Bax and caspase-8 [111].

In human hyperproliferative diseases, such as PAH and cancer, increased fission is associated with a low basal level of apoptosis. In PAH PASMC and cancer cells, MiD expression is increased, and Drp1-mediated mitotic fission is accelerated [20]. In PAH PASMC, silencing MiD49 or MiD51 increases apoptosis. Silencing MiDs also restored mitochondrial fusion and increased Bak expression while decreasing phosphorylation (activation) of the cell survival kinase, Akt, leading to increased apoptosis [20]. Thus, in human and experimental PAH, MiD upregulation, which is epigenetically regulated, confers apoptosis resistance [20]. Likewise, in human cancers, MiDs are epigenetically upregulated, and MiD knockdown increases mitochondrial fusion and triggers apoptosis and cell cycle arrest. Silencing MiDs in vivo regresses tumor growth in a xenotransplant non-small cell lung carcinoma model. The observed proapoptotic effect of MiD knockdown in A549 (lung cancer) and MCF7 (breast cancer) cells reflect increased expression of proapoptotic factors (Bax and Bak) and decreased expression of antiapoptotic mediators (Bcl-2 and Bcl-xL) [87]. MiD51 overexpression decreases the sensitivity of cells to apoptotic stimuli [88]. Consistent with this, HeLa cells lacking MiD51 are sensitized to different apoptotic stimuli [141]. Over expression of MiD51 did not induce the release of proapoptotic factors (cytochrome c, Smac/Diablo and AIF) from mitochondria. MiD51 overexpression and silencing led to a decrease and increase, respectively, of cleaved PARP in response to staurosporine treatment [88]. MiD51 deficiency promotes BCL2L1 dissociation from mitochondria which facilitates the recruitment of cytosolic Bax to the mitochondria. Consequent Bax homo-oligomerization on the OMM permeabilizes the membrane releasing SMAC Diablo and causing apoptosis [141]. Although most literature shows MiDs promote apoptosis resistance, Osellame et al. report the opposite, noting that loss of MiD51, MiD49 and MFF confers apoptosis resistance in MEF cells [97].

## 10. Mitochondrial Dynamics and Mitophagy

Mitophagy eliminates dysfunctional and damaged portions of mitochondria thereby maintaining a functional mitochondrial network [142]. There are two mitophagic pathways: ubiquitin-mediated mitophagy and receptor-mediated mitophagy [143]. In ubiquitin-mediated mitophagy, PINK1 is activated and recruited to the OMM, followed by the recruitment of the E3 ubiquitin ligase, Parkin, which polyubiquitinates several OMM proteins, including Mfn1 and Mfn2. These ubiquitinated proteins are then recognized by p62, optineurin (OPTN) and NDP52 which leads to recruitment of light chain 3 (LC3) and formation of the autophagosome [144].

Mfn2 deficiency, which contributes to mitochondrial fragmentation in PAH and lung cancer, results in part from increased proteasomal degradation triggered by PINK1-induced phosphorylation of S442 of Mfn2 [145]. Inhibiting either Mfn2 phosphorylation or the proteasome increased Mfn2 expression, enhances Mfn2’s fusogenic, antiproliferative and proapoptotic effects [145].

Receptor-mediated mitophagy relies on OMM proteins, such as FUDNC1, NIX, BNIP3, PHB2 and BCL2L13. These receptors contain an LC3 interacting region (LIR) motif that is recognized by LC3 and recruit mitochondria to the autophagosome [146,147]. However, Parkin-independent mitophagy pathways exist, and mitophagy can occur through other ubiquitin ligase and receptor-mediated mitophagy mechanisms or via a mechanism of lipid-mediated mitophagy [148]. Other E3 ligases implicated in the process of mitophagy include MUL1 [149], ARIH1 [150] and MARCH5 [151].

Mitochondrial membrane lipids also regulate mitophagy. Ceramide in the OMM interacts with LC3B-II to trigger mitophagy [152]. Cardiolipin translocation to the OMM also increases mitophagy. Externalized cardiolipin forms a platform on the OMM and binds LC3, which in turn initiates autophagosome formation and mitophagy [153]. Moreover, on the OMM, Beclin1, a mitophagy regulator, interacts with cardiolipin to promote mitophagy [154].

Miro1 (Mitochondrial Rho GTPase 1), which is involved in mitochondrial translocation, also participates in mitophagy. Parkin translocation to mitochondria promotes mitophagy, and Miro1 functions as a calcium-dependant docking site for inactive Parkin during mitochondrial damage and can recruit Parkin without stabilized Pink1 [155]. Knockout of Miro1 reduces Parkin translocation to the mitochondria and suppresses mitophagy [156]. These results indicate that Miro1 functions as a calcium-sensitive regulatory docking site for inactive Parkin on mitochondria.

## 11. Mitochondrial Dynamics and the Cell Cycle

Mitochondrial dynamics are regulated in parallel with cell cycle progression, thereby ensuring daughter cells have an equitable population of mitochondria. This mitochondrial coordination with mitosis is principally regulated by Drp1 [18]. Mitogen-induced activation of the cyclin D–CDK4/6 complex in early G1 phase, triggers phosphorylation and inactivation of pocket proteins (pRB, p107 and p130) (Figure 10). This releases E2F transcription factors, which modulate the expression of genes necessary for the lipid, protein and nucleic acid synthesis that supports cellular proliferation [157]. Non-phosphorylated pocket proteins, when complexed to E2F proteins, repress genes necessary for cellular proliferation, such as cyclin E. This suppression ceases when the cyclin D–CDK4/6 complex in early G1 phase phosphorylate and inactivate the pocket proteins [157] (Figure 10 and Figure 11). In proliferating cells, mitochondria undergo hyperfusion during the G1 to S transition and mitochondria form a “giant, single, tubular network at G1/S transition” [158] (Figure 10). During the S phase, CDK2 becomes active and complexes with cyclin E. During the G2 phase, CDK2 complexes with cyclin A [157]. Finally, CDK1 complexes with cyclin B to phosphorylate Drp1 at S616 and mediates mitochondrial hyperfission, which is required to allow cells to complete mitosis [64] (Figure 10).

At prophase, the first mitotic stage, mitochondria prepare for fission by recruiting Drp1 at the OMM through interactions with its receptors (MFF, Fis1, MiD49 and MiD51). During prophase, and peaking in metaphase, Drp1-induced fission causes a highly fragmented mitochondrial network (Figure 11) [159,160]. As sister chromatids separate from each other during anaphase, mitochondria remain fragmented. During telophase, mitochondria begin to form elongated structures through the tandem actions of pro-fusion proteins Mfn1/2 at the OMM and OPA1 at the IMM. Upon completion of cytokinesis, mitochondria form a hyperfused, interconnected network within each daughter cell (Figure 11) [159]. At the end of the M phase, likely during anaphase, a fused mitochondrial network is re-established due to the proteasomal degradation of Drp1, mediated by the anaphase-promoting complex (APC) [161] (Figure 10). Thus, changes in mitochondrial morphology observed in mitosis such as induction and resolution of fission are related to Drp1 activation and are substantially mediated by CDK1 and the APC, respectively [157] (Figure 10 and Figure 11). Inhibiting Drp1 or MiDs causes cell cycle blockade in PASMC, lung and breast cancer cell lines [20,133].

Cell proliferation generates high energy demands and the bioenergetic activity of mitochondria is crucial for cell cycle progression [157]. Under conditions in which there is metabolic impairment and decreased ATP levels, the resulting rise in the AMP/ATP ratio activates AMPK and leads to G1-S arrest [162]. Mitochondrial electron transport chain (ETC) defects, such as loss-of-function mutants in the ETC complexes I or IV, induce G1-S arrest [163]. Mitochondrial hyperfusion early in the cell cycle is required for Cyclin E accumulation, which in turn is needed for entry into S phase. Triggering mitochondrial hyperfusion in serum starved cells causes entry into S phase [158].

In lung cancer cells, interventions that sustain mitochondrial fusion (whether caused by inhibiting Drp1 or augmentation of Mfn2) reduce cell proliferation and increase spontaneous apoptosis [133]. The increased MiD expression in PAH PASMC also accelerates Drp1-mediated mitotic fission and increases cell proliferation. Silencing MiDs promotes mitochondrial fusion and causes cell cycle arrest at the G1 phase in PAH PASMC [20]. Thus, inhibition of Drp1 and/or the MiDs is an attractive therapeutic strategy for preventing pathologic cell proliferation.

The antiproliferative effect of Mfn2 is mediated by its ability to bind Ras in its effector binding region, which inhibits the Ras–Raf–ERK signalling pathway. In Mfn2 knockout MEF’s two-fold increase in cell proliferation was observed compared to wild type cells [34]. Furthermore, downregulation of Mfn2 in peripheral blood T cells preceded the cells’ entry into the cell cycle, suggesting that downregulation of Mfn2 is critical for allowing activated T-cells to enter the cell cycle. In addition, inhibiting mTORC1 by rapamycin blocked Mfn2 downregulation and prevented cell entry into the cell cycle, indicating an important role of the mTOR pathway in the activation-induced downregulation of Mfn2 in peripheral human T-cells [35].

While the main mitochondrial mediators of cell cycle regulation appear to be Drp1 and Mfn2, Fis1 also interacts with components of the cell cycle machinery at the G2–M transition [164]. Fis1 depletion caused fusion and impaired cell cycle progression through G2–M phase, which was rescued by the reintroduction of Fis1 [164]. Additionally, in Fis1 knockdown cells, the expression of cell cycle regulators governing the G2–M phase, including cyclin A, cyclin B1, CDK1, polo-like kinase1 (PLK1), aurora kinase A and MAD2, were suppressed 2–10-fold [164]. Thus, it appears that fusion in early mitosis (mediated by mitofusins) is necessary to support the bioenergetics of cell division, while fission (mediated by Drp1, MiD49, MiD51 and perhaps Fis1) is required subsequently to allow mitochondrial distribution to daughter cells. Thus, a mitochondrial dynamics cycle runs in parallel with the nuclear division cycle, which has classically been the focus of studies of cell proliferation.

## 12. Mitochondrial Dynamics and Cancer

### 12.1. Lung Cancer

Abnormal mitochondrial dynamics in lung cancer influence disease pathogenesis [133]. Non-small cell lung cancer (NSCLC) tissues in microarrays have reduced Mfn2 and increased Drp1 expression, relative to adjacent noncancerous tissue [133]. Likewise, in lung cancer cell lines (H358, A549 and H1993), mitochondrial fission is increased due to the combined effects of increased Drp1 and decreased Mfn2 expression [133]. Activated Drp1 (both phosphorylated at serine 616 and/or dephosphorylated at serine 637) is also increased in NSCLC lines [133]. Inhibition of Drp1, achieved using mdivi-1, or overexpression of Mfn2, using an adenoviral vector, caused mitochondrial hyperfusion in NSCLC, decreased cell proliferation rates and increased apoptosis [133]. Moreover, in a mouse xenograft NSCLC model, treatment with Mdivi-1 or overexpression of Mfn2 regressed tumor size. This indicates that abnormal mitochondrial dynamics favoring fission, promote growth of NSCLC tumors through a Drp1/Mfn2 imbalance that increases mitotic fission and accelerates cell cycle progression [133].

It is not just increased Drp1 activation that promotes fission and cell proliferation in hyperproliferative diseases. MiDs are also pathologically elevated in NSCLC through an epigenetic mechanism (decreased miroRNA-34a-3p expression) [87] Silencing of MiDs caused mitochondrial fusion, triggered cell cycle arrest and increased apoptosis. Inhibiting MiD expression in vivo also regressed tumors in a xenotransplant NSCLC model [87]. The expression of MFF, but not Fis1, was also increased in these NSCLC cells; however, inhibiting MFF and Fis1 did not inhibit mitochondrial fission. Thus, MiD49 and MiD51, as well as Drp1, played the predominant role in NSCLC progression and are likely therapeutic targets.

Consequently, we initiated a program to discover novel, small molecule Drp1 inhibitors, which led to the patenting of Drpitor1 and Drpitor1a [23]. Through in silico screening of a small molecule library against the crystal structure of Drp1, targeting the GTPase domain, we identified an ellipticine compound, Drpitor1, which is a specific, potent, Drp1 inhibitor. This compound, along with a Drpitor1 congener lacking the methoxymethyl group, called Drpitor1a, inhibits Drp1 GTPase activity without affecting the GTPase activity of Dynamin 1 [23]. Drpitor1a is approximately 50-fold more potent than mdivi-1 [23], a widely used Drp1 inhibitor. The half maximal inhibitory concentrations of mitochondrial fragmentation achieved using Mdivi-1, Drpitor1 and Drpitor1a in A549 cancer cells was 10, 0.090 and 0.057 μM, respectively, showing the higher potency of Drpitor1 and Drpitor1a [23]. We also showed that Dpritor1a directly inhibited Drp1 that was isolated from cancer cells without inhibiting dynamin 1, another large GTPase [23]. Both Drpitors reduce proliferation and induce apoptosis in cancer cells and regressed NSCLC tumors in a mouse xenotransplantation model [23].

While most of the literature indicates that there is increased fission in NSCLC, agreeing with the original reports from our lab, one report indicates that fission is decreased in NSCLC through a mechanism dependent on the tumor suppressor, SIRT4 [165]. SIRT4 inhibited Drp1 phosphorylation and decreased Fis1-dependent Drp1 recruitment to the OMM [165]. However, in areas with low SIRT4 expression, high levels of Drp1 phosphorylation and increased fission were observed [165].

### 12.2. Hepatocellular Cancer

Mitochondrial fission promotes cell migration, autophagy, tumor-associated macrophage infiltration and tumor progression in hepatocellular carcinoma (HCC) [166,167]. Fission was upregulated in HCC due to elevated Drp1 and depressed Mfn1 expression [168], which is consistent with findings in lung cancer [133], albeit in HCC the dysregulated mitofusin was Mfn1. In HCC tissues vs. adjacent non-tumor liver tissues, transmission electron microscopy showed reduced mitochondrial length, indicating increased mitochondrial fission [168]. HCC patients with high Drp1 expression, low Mfn1 expression or a high ratio of Drp1:Mfn1 had a significantly lower overall survival rate than those with a low Drp1:Mfn1 ratio [168], suggesting mitochondrial dynamics are potential cancer biomarkers.

Mutations of the tumor suppressor *TP53* are important in HCC and contribute to mitochondrial fission. p53 inhibits mTORC1 signaling and reduces expression of mitochondrial fission process 1 (MTFP1), which promotes phosphorylation and activation of Drp1 [169]. Huang studied the effects of *TP53* mutations on mitochondrial dynamics and cell survival in HCC cell lines. Compared to wild type, *TP53* cells with TP53 loss of function, point mutations had increased mitochondrial fragmentation and activation of extracellular signal-regulated kinase 1/2 (ERK1/2) signaling, resulting in epithelial-to-mesenchymal transition and invasive cell migration [168]. Moreover, they found that increased mitochondrial fission promoted the survival of HCC cells by facilitating autophagy and inhibiting apoptosis. Huang further proposed that fission-induced HCC cell survival was mediated by elevated ROS production and AKT-mediated phosphorylation of E3 ubiquitin protein ligase, facilitating TP53 ubiquitination and degradation. This study also provided evidence that mitochondrial fission leads to AKT-mediated activation of NFKB. Huang et al., also found that Drp1 knockdown induced significant depolarization of mitochondrial membrane potential whilst Drp1 overexpression hyperpolarized mitochondria [168]. Drp1 knockdown decreased, whilst overexpression increased, oxygen consumption rates [168]. Our group has reported mitochondrial hyperpolarization as a hallmark of both PAH and cancer cells (A549, M059K and MCF-7) [170,171].

Consistent with the adverse role of excess fission in HCC, the observed depression of Mfn1 triggers the epithelial-to-mesenchymal transition (EMT) of HCC cells [172]. EMT, which disrupts cell–cell and cell–matrix adhesion, is associated with a more aggressive migratory and invasive cancer cell phenotype and, in HCC, is associated with metastasis and poor prognosis [173].

While the preponderance of the literature suggests increased Drp1 fission or impaired Mfn activity promotes HCC progression, Li et al. made a conflicting report, suggesting that metabolic reprogramming via mitochondrial elongation was essential to HCC cell survival [174]. They further reported increased mitochondrial fusion in HCC patient tissues and in vivo culture tumor organoids [175]. In their study, knockdown of the fusion mediators OPA1 or Mfn, inhibited fusion in HCC cell lines and reduced cell growth in vitro by inducing apoptosis, leading to reduced tumor growth in vivo [175]. Li et al. used transcriptomic profiling to show that that fusion inhibition altered metabolic pathways, reducing oxygen consumption and cellular ATP production. They concluded, that increased mitochondrial fusion in liver cancer alters metabolism and promotes tumor cell growth [175].

### 12.3. Ovarian Cancer

Drp1 also contributes to the regulation of mitosis and the development of primary and relapsed epithelial ovarian cancer (EOC) [176]. Drp1 co-expresses with the cell cycle module responsible for mitotic transition in EOC and inhibition of Drp1 attenuated mitotic transition [176]. Tanwar et al. also reported that Drp1 co-expression with cell cycle mediators identifies primary EOCs that are universally responsive to chemotherapy. They further suggested that Drp1-driven mitosis may underlie sensitivity of primary EOC tumors to chemotherapy [176]. Drp1 (and MFF) expression in EOC correlates with cell cycle genes; specifically, those which promote the mitotic transition of the cell cycle [176]. This is consistent with previous findings which indicate that Drp1 activity/expression are elevated during mitotic transition of the cell cycle [64,87,133]. Downregulation of Drp1, in the A2780 EOC cell line, inhibited mitotic transition, resulting in G2–M phase cell cycle arrest and slowed cell proliferation [176], consistent with the observation that the Drp1 inhibitor mdivi-1 causes G2-M arrest in lung cancer cells [133].

Han et al. showed that hypoxia causes mitochondrial fission and promotes cisplatin resistance in ovarian cancer cells by induced reactive oxygen species (ROS) production [177]. These effects in ovarian cancer cells are mediated by downregulation of p-Drp1 (Ser637) and Mfn1 [177]. Inhibition of Drp1, using Mdivi-1 or siDrp1, restored cisplatin sensitivity in hypoxic ovarian cancer cells, indicating that inhibiting fission could be a mitochondrial-targeted anticancer therapy in ovarian cancer [177], as originally proposed in lung cancer [133].

Metabolic reprogramming, the Warburg effect, is characterized by increased aerobic glycolysis and decreased oxidative phosphorylation. This reprogramming uses accelerated rates of uncoupled glycolysis to sustain cellular bioenergetics, while reducing mitochondrial-mediated apoptosis, thus supporting cellular growth in various cancer cells and PAH PASMC [178]. Metabolic programming in ovarian cancer cells is influenced by salt-inducible kinase 2 (SIK2), a member of the AMPK family [179]. SIK2 promotes the growth of ovarian cancer cells by increasing cell proliferation and EMT while inhibiting apoptosis [179]. SIK2 induces the Warburg effect in ovarian cancer cells by upregulating HIF-1a expression, through activation of the PI3K/AKT signaling pathway. SIK2 also promotes mitochondrial fission through phosphorylation of Drp1 at Ser616, which also inhibits mitochondrial oxidative phosphorylation (Gao et al., 2020). Thus, SIK2 impairs glucose metabolism and increases mitochondrial fission in ovarian cancer cells by activating the PI3K/AKT/HIF-1a pathway [179].

### 12.4. Breast Cancer

Dasgupta et al. assessed the role of MiD49 and MiD51 as drivers of cellular proliferation and apoptosis-resistance in IBC [87]. MiD-levels were pathologically upregulated in IBC through an epigenetic mechanism: specifically, the decreased expression of microRNA-34a-3p, a negative regulator of both MiD 49 and MiD51 [87]. MiD silencing caused cell cycle arrest by increasing expression of cyclin inhibitors (p27 and p21), inhibition of Drp1 and inhibition of the AKT–mTOR–p70S6k pathway [87]. In addition, there was a positive correlation between MiD expression and tumor size and grade in breast cancer patients [87].

Mitochondrial fission promotes apoptosis resistance, drug resistance and increases the survival of breast cancer cells [180]. Mitochondrial fission was significantly increased in breast cancer tissues, particularly tissue from triple negative breast cancer (TNBC) patients [181]. Drp1 was upregulated and Mfn1 downregulated in TNBC, and the resulting increase in mitochondrial fission predicted poor patient prognosis due to enhanced cancer cell survival [181]

A positive feedback loop exists between mitochondrial fission and Notch signaling pathways in TNBC cells. Activation of Notch signaling enhances Drp1-mediated mitochondrial fission, while Drp1 activation promotes Notch signaling; a cycle which culminates in increased survivin expression [181]. Inhibition of Drp1 or Notch1 significantly impairs the activation of the other, decreasing survivin levels and reducing TNBC cell survival and proliferation [181].

The localization and distribution of mitochondria has a close association with fission and fusion dynamics, and pathological mitochondrial migration has been implicated in MDA-MB-231 breast epithelial carcinoma cells [182]. In these cells, the anterior localization of mitochondria in between the nucleus and the leading edge of migrating epithelial cancer cells correlates with increased migration velocity and invasive metastasis [182]. Stimulation of fusion, through augmentation of OPA1, or inhibition of fission, by administration of a dominant negative Drp1 mutant, decreased mitochondrial fission. The resulting larger mitochondria are less efficiently transported, and their distribution is more homogenous, limiting anterior mitochondrial localization and decreasing cancer cell migration rates [182]. Additionally, interfering with the expression of Miro 1, which links mitochondria to microtubules and facilitates transport, reduces the migratory capabilities of cancer cells. These findings underscore the importance of another noncanonical mitochondrial dynamics function, mitochondrial localization and translocation, on the invasiveness of epithelial cancer cells. Drp1 expression is upregulated in human breast carcinomas, and metastases and mitochondria are most fragmented in metastatic cell lines [183]. Silencing of Drp1 or overexpression of Mfn2 inhibited lamellipodia formation and decreased rates of cell migration and invasion in breast cancer cells. Thus, abnormal mitochondrial dynamics have a central role in regulating cancer cell migration and invasion, perhaps by redistributing mitochondria to areas of higher energy demand [183], in addition to regulating the cell cycle and apoptosis [87].

Metastatic breast cancer cells are known to preferentially migrate to organs with a soft extracellular matrix microenvironment [184]. Romani et al., report that cancer cells cultured on a soft extracellular matrix display increased mitochondrial F actin association, mediated by Spire 1C and Arp2/3 nucleation factors and display increased Drp1- and MiD49/51-dependent mitochondrial fission. Matrix-induced changes in mitochondrial dynamics increase ROS, cysteine uptake and glutathione metabolism, creating resistance to oxidative stress and ROS-dependent chemotherapy drugs in these cells. Ramani concluded that a soft extracellular matrix adversely alters redox homeostasis in metastatic breast cancer cells and that targeting this mechanotransduction pathway using Drp1 inhibitors, such as Drpitor1a, may constitute a new approach to prevent chemotherapy resistance [184]. Thus, in breast cancer Drp1-dependent fission is increased which supports cancer cell survival and metastatic potential by inhibiting apoptosis, accelerating cell cycle progression, increasing cell migration, increasing ROS generation and impairing bioenergetics, as occurs in most cancers (Figure 12).

## 13. Mitochondrial Dynamics and Pulmonary Arterial Hypertension (PAH)

PAH is characterized by the obstruction of small pulmonary arteries due, in part, to excessive proliferation and an apoptosis-resistant phenotype in vascular cells (smooth muscle cells, endothelial cells and fibroblasts). This vascular obstruction, together with inflammation, thrombosis and vasoconstriction, contributes to the pathogenesis of PAH. Increasing afterload faced by the right ventricle (RV), and RV inflammation mediated by activation of the NLRP3 inflammasome [185], leads to RV failure and a 50% mortality rate at 5 years post diagnosis [186]. This pseudoneoplastic, hyperproliferative, obstructive, vasculopathy of PAH shares several mitochondrial characteristics with cancer; specifically, a shift to aerobic glycolysis, the Warburg phenomenon [187], and increased mitochondrial fragmentation, due to increased fission [21] and impaired fusion [186] (Figure 13).

## 14. Mitochondrial Fission and PAH

Increased mitochondrial fragmentation in PAH reflects increased Drp-1-mediated mitotic fission [21] (Figure 13). The molecular basis of this structural change in mitochondria in PAH PASMC includes upregulation of cyclin B1 CDK1, phosphorylation and activation of Drp-1 at serine 616 [21]. Calcium-sensitive calmodulin kinase (CaM kinase) may phosphorylate Drp-1 and contribute to Drp1 activation [186]; however, in PAH PASMCs a CDK1 inhibitor (RO-3396) was more effective than a CamK inhibitor in reducing Drp1 phosphorylation, which suggests that cyclin B-CDK1 is the predominant mediator of Drp1 activation in PAH [21].

Preventing mitochondrial fission by inhibiting Drp1 has therapeutic potential for PAH patients. Mdivi-1, which inhibits the GTPase activity of Dnm1 (a yeast homolog of Drp1) with an IC50 of 1–10 μM, inhibits staurosporine-induced mitochondrial fission in mammalian COS cells [188]. However, Bordt et al. subsequently reported that Mdivi-1 was a poor inhibitor of Drp1 GTPase and instead primarily acted by inhibiting mitochondrial complex I, thereby changing mitochondrial ROS production [189]. They also suggested that Mdivi-1 is likely to have multiple cellular targets that are independent of Drp1 due to its thiophenol molecular structure [189]. In our experience Mdivi-1 inhibits Drp1-mediated fission, regresses obstructive pulmonary vascular remodeling and improves hemodynamics in vivo in an animal model of PAH [21]. Mdivi-1 is also cardioprotective, inhibiting mitochondrial fission in rat, left ventricular, cardiomyocytes and improving left ventricular function ex vivo, in a rat Langendorff ischemia-reperfusion injury model, and in vivo, in a mouse cardiac arrest model [22]. Likewise, a 7 amino acid competitive peptide, P110, is a proposed inhibitor of Drp1–Fis1 interaction [190] that selectively inhibits Fis1-mediated mitochondrial fission [191]. In an ex vivo right ventricle (RV) Langendorff, ischemia-reperfusion injury model, P110 improves mitochondrial function and preserves RV diastolic function in the RV of normal rats and rats with monocrotaline (MCT)-induced PAH [82]. MCT (60 mg/kg) by subcutaneous injection is a well-established model of PAH due to the predisposition of the MCT RV to develop fibrosis and the MCT rat to die of RV failure: MCT models of PAH also display a fragmented mitochondria due to excessive Drp1 activation [192].

RV fibrosis is an important contributor to RV failure in PAH. Tian et al. studied mitochondrial metabolism and fission in RV fibroblasts in human and experimental PAH. Fibroblasts from PAH RVs are rapidly proliferating, have a Warburg metabolic phenotype and increased Drp1-dependent mitochondrial fission. Inhibiting Drp1 prevents mitochondrial fission and reduces RV fibroblast proliferation and collagen production [192]. Thus, inhibiting the excess fission seen in the pulmonary vasculature and RV in PAH has the potential to regress pulmonary vascular disease, reduce RV fibrosis and improve RV function.

MiD49 and MiD51 are also pathologically upregulated in PAH. Increased MiD expression contributes to mitochondrial hyper fragmentation and the hyperproliferative, apoptosis-resistant phenotype of PAH PASMC’s [20]. Inhibition of MiDs by specific siRNAs restores mitochondrial morphology (Figure 8) and arrests cell cycle progression at the G1 phase in PAH PASMC. Moreover, heterologous overexpression of MiDs in normal PASMC increases mitochondrial fission sufficiently to accelerate cellular proliferation, recapitulating the PAH phenotype. MiDs regulate cell proliferation through ERK1/2 and cyclin-dependent kinase 4 (CDK4)-dependent mechanisms [20]. In rodent models of PAH silencing MiDs or augmenting their regulator miR (miR-34a-3p) regresses experimental PAH [20].

## 15. Mitochondrial Fusion and PAH

Downregulation of Mfn2 expression contributes to the observed mitochondrial fragmentation seen in PAH PASMC. Transfection of normal PASMC with siRNA targeting Mfn2 recapitulates a PAH phenotype, leading to a fragmented mitochondrial network and increased rates of PASMC proliferation [52]. Mfn2 downregulation is recapitulated in rodent models of PAH created in female rats exposed to SU5416 plus chronic hypoxia (CH + SU) [52]. The CH + SU model involves the subcutaneous implantation of Sugen 5416 (semaxinib) combined with exposure to three weeks of chronic hypoxia (10% O2) followed by 2 weeks of re-exposure to normoxia [52]. Five weeks following this procedure, this “2-hit” model of CH + SU develops sever progressive PAH [193]. Several proposed mechanisms may explain how Mfn2 downregulation is acquired in PAH. Mitogens which are upregulated in PAH, such as PDGF and endothelin 1 (ET-1), can reduce Mfn2 expression in systemic arterial SMC [38]. Downregulation of Mfn2 may also be connected to the observed downregulation of PGC-1a, a transcriptional co-activator of Mfn2, in PAH [52]. Hypoxia (and presumably the pseudohypoxic state of PAH) reduces Mfn2 expression in normal PASMC by ~20% [52]. Overexpression of Mfn2, using an adenoviral vector, restores Mfn2 expression and fuses the mitochondrial networks in Mfn^−^/^−^ MEF cells and in human PAH PASMCs. This therapeutic increase in Mfn2 expression, reduces cellular proliferation and restores normal rates of apoptosis in PAH PASMC [186]. In vivo overexpression of Mfn2, through airway nebulization of an adenoviral Mfn2 vector, regresses PAH in the chronic hypoxia–SU5416 PAH model in female rats, as judged by improved treadmill walking distance, decreased pulmonary vascular resistance and decreased PA medial thickness. The similar inhibitory effects of Drp-1 inhibition and Mfn2 augmentation on hyperproliferation suggest that achieving network fusion is important to the anti-proliferative, pro-apoptotic effects of both Drp-1 inhibition and Mfn upregulation [21,186]. In addition, Mfn2 is phosphorylated at serine 442 by PINK1 in PAH, leading to its ubiquitin-mediated proteasomal degradation and increased fission [145].

## 16. Conclusions and Perspectives

Considerable progress has been made in identifying the normal mechanisms of mitochondrial dynamics in health. Questions remain regarding normal mitochondrial fission, such as the importance and mechanism underlying central versus peripheral mitochondrial fission and the role of DNM2 as a putative terminal step in fission. Acquired disorders of mitochondrial dynamics participate in common diseases, such as PAH and most cancers. Mitochondria regulate cell proliferation, through the role of mitotic fission in cell cycle regulation, in addition to regulating apoptosis. It is widely accepted that post-translational activation of Drp1 and/or downregulation of Mfn2 are major drivers of the proliferation/apoptosis imbalance that characterizes PAH and many cancers. The field has been advanced by the ability to inhibit mitochondrial fission using small molecule GTPase inhibitors, such as Mdivi-1 [188] and Drpitor1a [23]; however, drug specificity concerns remain, particularly for Mdivi-1. In PAH and cancer, increased fission, due to upregulation and activation of Drp1, and decreased fusion, due to downregulation of Mfn2, contribute to disease pathophysiology and can be therapeutically targeted.

## Figures and Tables

**Figure 1 cells-12-01897-f001:**
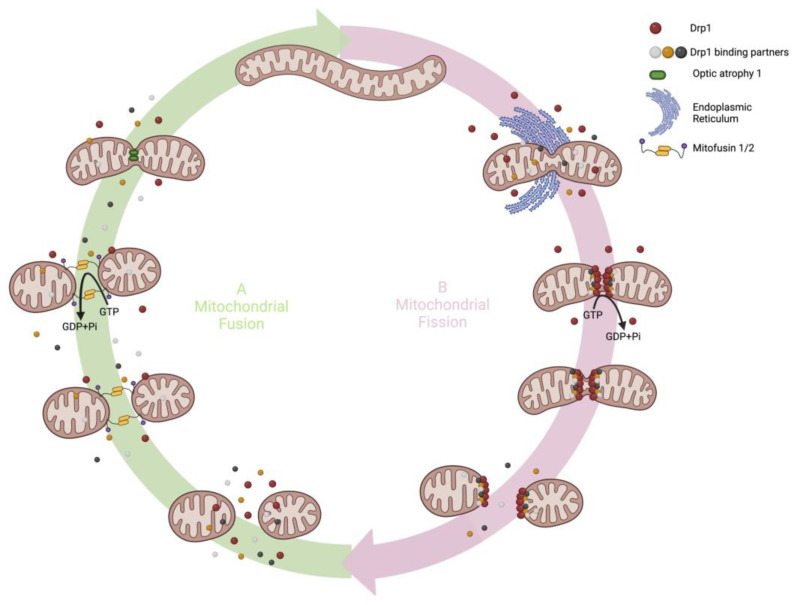
A simplified scheme of mitochondrial fission and fusion in mammalian cells. (**A**) Schematic representation of fusion. The outer membranes of two adjacent mitochondria are tethered by the interaction in trans of the HR2 domains of Mfns. GTP binding and hydrolysis cause Mfns conformational change leading to OMM fusion. Following OMM fusion, OPA1 drives IMM fission. (**B**) Schematic representation of fission. Fission is initiated by ER-mediated constriction and marks the site for further constriction by Drp1 and its binding partners. Drp1 is recruited from the cytosol to the fission site via its receptors (MFF, MiD49 and MiD51) and forms multimeric contractile rings at the fission site. GTP hydrolysis powers Drp1 multimers to make conformational changes that constrict and divide the organelle.

**Figure 2 cells-12-01897-f002:**
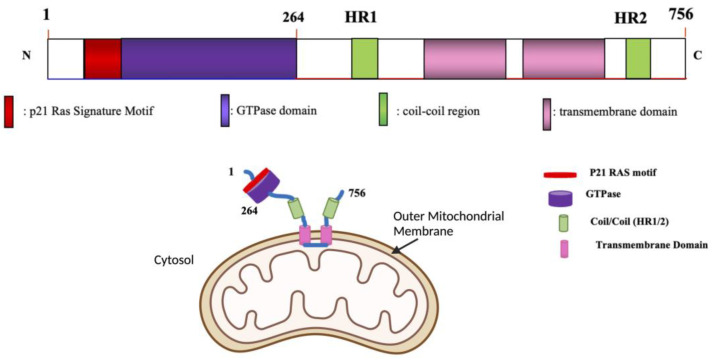
Schematic representation of Mfn2 protein showing the structural domains of this fusogenic large GTPase. N = (amino terminus), C = (Carboxyl Terminus). Adapted with permission from Chen and Dasgupta et al. [34,35].

**Figure 3 cells-12-01897-f003:**
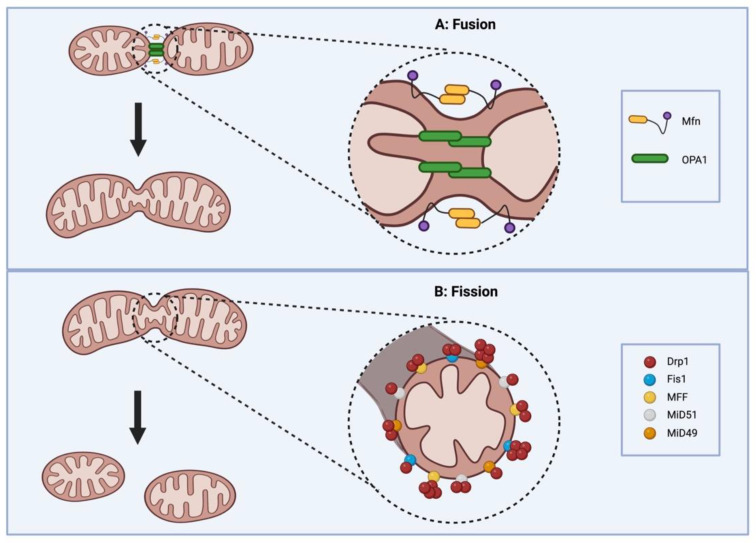
Mitochondrial dynamics. (**A**) Shows joining of mitochondria by the process of fusion, which is mediated by the OMM fusion protein isoforms, Mfn1 and Mfn2, and the inner mitochondrial (IMM) fusion protein OPA1 (labelled in yellow and green respectively). Fusion causes the intermixing of the contents of the mitochondrial matrix of two discrete mitochondria. This process of complementation results in a healthier, elongated, single mitochondrion. (**B**) Shows the division of mitochondria by the process of fission. Fission is mediated by Drp1, which is recruited to the mitochondria from the cytosol after post-translational activation, often by phosphorylation at serine 616. OMM adapter proteins, such as MFF, Fis1, MiD49 and MiD51, focus on multimeric Drp1 assembly. A Drp1-enriched fission ring surrounds the mitochondria forming a ring-like structure which cleaves the organelle causing fission.

**Figure 4 cells-12-01897-f004:**
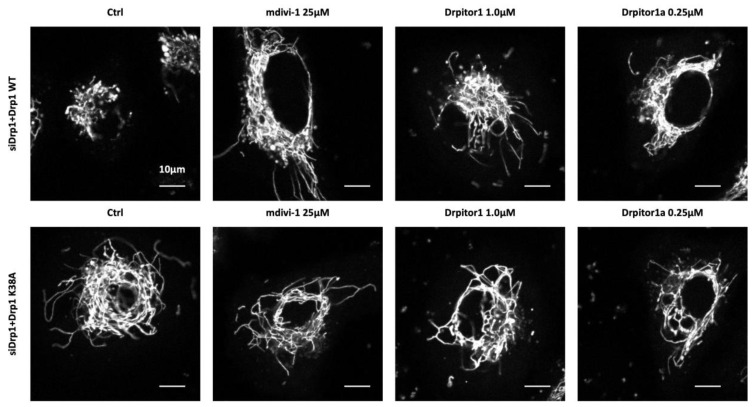
Confocal Examples of Drp1 Inhibition. Representative images (top row) show that Mdivi-1 and Drpitor1/1a both fuse the mitochondrial network in A549 cells, in which endogenous Drp1 has been knocked down by siRNA and replaced by Drp1 transfection. In the lower panel, confocal microscopy shows cells in which endogenous Drp1 has been knocked down but replaced with a loss-of-function Drp1 mutant (Drp1 K38A). This nonfunctional Drp1 results in a fused mitochondrial network, and there is no effect on this when the cells are exposed to Mdivi-1 (25 μM), Drpitor1 (1.0 μM) or Drpitor1a (0.25 μM). Reproduced with permission from Wu et al. [23].

**Figure 5 cells-12-01897-f005:**
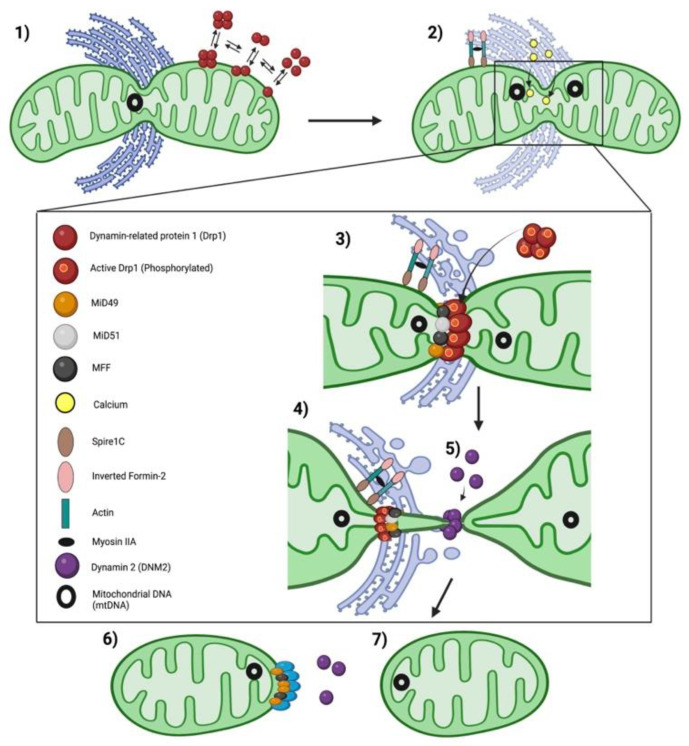
Proposed model of Mitochondrial fission in mammals. (**1**) Mitochondrial DNA (mtDNA) replication marks the site for the recruitment of the endoplasmic reticulum (ER) to contact the mitochondria at a site of impending midzone fission [70]. In parallel, Drp1 oligomers are in constant balance between the cytosol and mitochondria, moving to the OMM when post-translationally activated. (**2**) At these sites, the ER-bound inverted formin-2 (INF2) and mitochondrial-bound Spire1C establish actin cables between the two organelles, with myosin II allowing actin contraction, to provide the mechanical force to induce mitochondrial preconstriction. ER-Ca^2+^ release into mitochondria, via mitochondrial calcium uniporter, leads to inner mitochondrial membrane (IMM) constriction upstream of dynamin-related protein-1 (Drp1) recruitment [71]. (**3**) At mitochondrial preconstriction sites, adaptors of Drp1, MiD49/51 and mitochondrial fission factor (MFF) accumulate and recruit activated Drp1, which oligomerizes in a ring-like structure around the mitochondrial tubule. (**4**) Upon GTP hydrolysis, Drp1 changes conformation and increases mitochondrial constriction. (**5**) DNM2 is recruited to Drp1-mediated constriction sites where it assembles and finishes membrane scission (**6**) leading to two daughter mitochondria. It is uncertain whether DNM2 is required to complete fission. (**7**) Drp1 appears to be concentrated on one daughter mitochondria following fission.

**Figure 6 cells-12-01897-f006:**
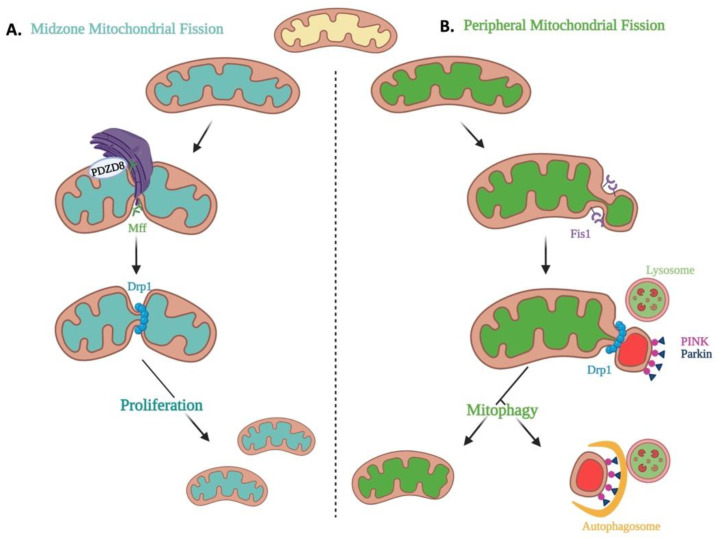
Mitochondrial Midzone and Peripheral Fission. (**A**) Shows the fate of dividing mitochondria through the process of midzone mitochondrial fission (defined as fission occurring within the central 50% of a mitochondrion). Midzone fission occurs at sites guided by actin and PDZD8 mitochondrial–ER tethering, at areas of endoplasmic reticulum pre-constriction. Midzone fission is proposed to rely predominantly on MFF and Drp1 assembly. Midzone fission appears to support cell proliferation and mitochondrial biogenesis. (**B**) Shows the fate of dividing mitochondria through peripheral fission events (defined as fission occurring at the peripheral 0–25% of the mitochondrion). Peripheral fission is also mediated by Drp1; however, this process appears to rely on Fis1 and lysosomal contact to mediate Drp1 assembly. Division at the periphery enables damaged material to be shed into smaller mitochondria destined for mitophagy, an important mechanism for mitochondrial quality control.

**Figure 7 cells-12-01897-f007:**
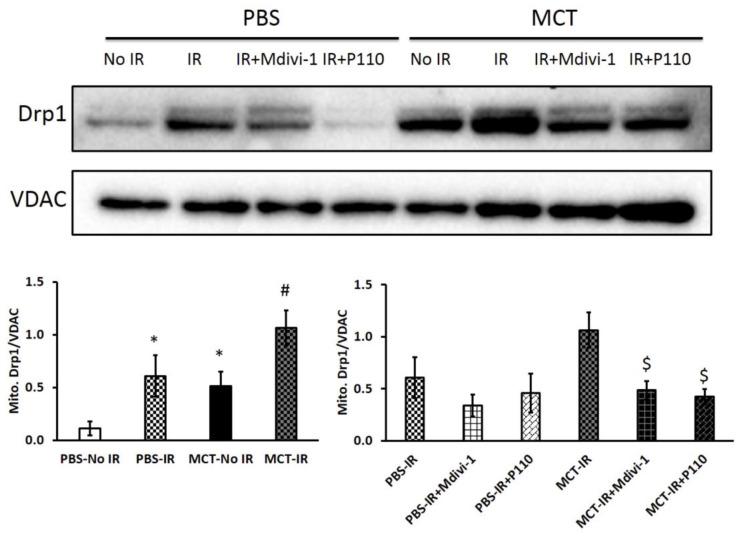
Mdivi-1 and P110 decrease mitochondrial Drp1 expression in both control and PAH right ventricles (RVs) following ischemia reperfusion injury (IR). Immunoblots of RV tissue show translocation of Drp1 to the mitochondria was increased in MCT versus control RVs (PBS = saline). At baseline (before IR) there is increased mitochondrial Drp1 in MCT RVs, indicating elevated basal rates of fission. Mitochondrial Drp1 expression increased in both PBS and MCT RVs in an ex vivo RV Langendorff, ischemia-reperfusion injury model. Both Mdivi-1 and P110, a proposed peptide inhibitor of the interaction between Fis1 and Drp1, reduced the translocation of Drp1 to the mitochondria, though only significantly in MCT RVs with IR. n = 3~5/group. MCT, monocrotaline. * *p* < 0.05 vs. PBS-No IR; # *p* < 0.05 vs. MCT-No IR; $ *p* < 0.05 vs. MCT-IR. Reproduced with permission from Tian et al. [82].

**Figure 8 cells-12-01897-f008:**
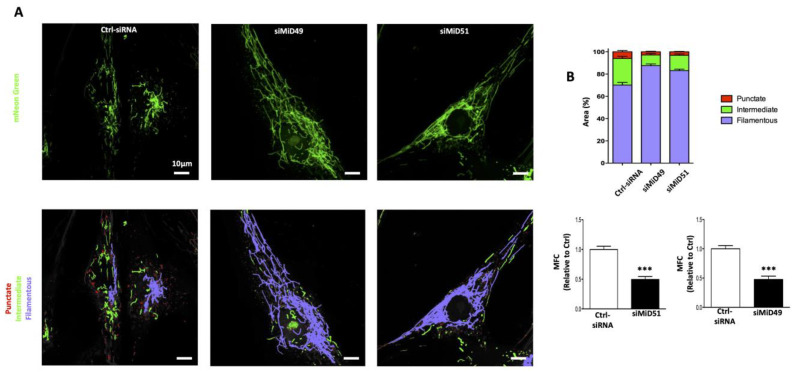
Silencing of MiD49 and MiD51 promotes mitochondrial fusion in PAH PASMC. (**A**) Mitochondrial fragmentation, evident in PAH PASMC, is reversed by silencing of MiD49 or MiD51. Representative images of mitochondrial networks in PAH PASMC. PAH PASMC were transfected with the specified siRNA, infected with Adv-mNeon Green and imaged after 48 h following infection. Mitochondria were color coded by their morphology using the process of machine learning (Wu et al., 2020): red: punctate; green: intermediate; and purple: filamentous. Scale bar: 10 μm. (**B**) Silencing of MiD49 or MiD51 reduces mitochondrial fission. Mitochondrial fragmentation was quantified by mitochondrial fragmentation count (MFC) and through machine learning, which automates the measurement of percentage area of punctate, intermediate and filamentous mitochondria of each image. (***) = *p* < 0.001. Adapted with permission from [20].

**Figure 9 cells-12-01897-f009:**
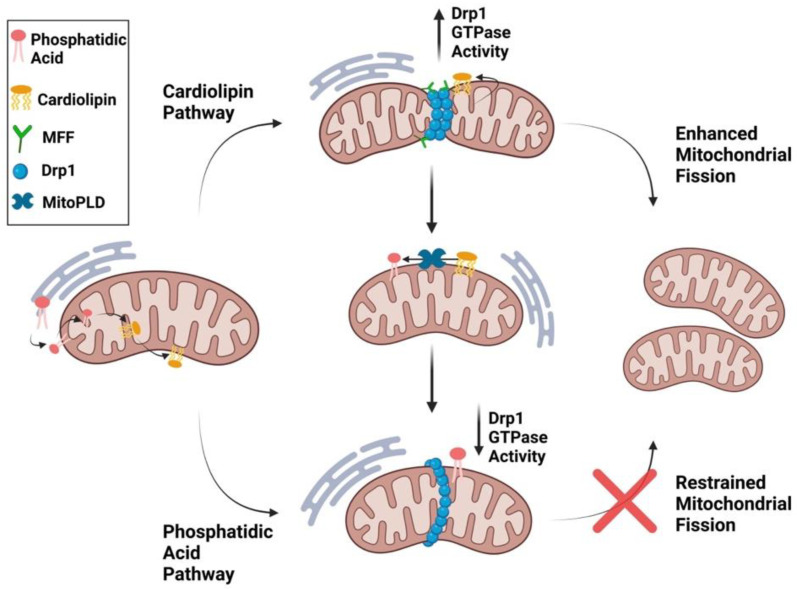
Regulation of Mitochondrial Division by Phosphatidic Acid and Cardiolipin. Phosphatidic acid (PA) is synthesized in the endoplasmic reticulum and transported to the mitochondrial outer membrane. A fraction of PA is transported to the inner mitochondrial membrane where it is converted to cardiolipin through cardiolipin synthase. Cardiolipin synthesized in the IMM is transported to the OMM where it interacts directly with Drp1 through its variable domain. This interaction stimulates Drp1 oligomerization and GTPase activity thereby enhancing fission. Cardiolipin-stimulated Drp1 GTPase activation is synergistically enhanced by MFF, suggesting that MFF acts with cardiolipin to potentiate mitochondrial fission. At the OMM, cardiolipin can be transformed back into phosphatidic acid by the mitochondrial OMM-localized enzyme, phospholipase D (MitoPLD). Phosphatidic acid inhibits Drp1-oligomerization-induced GTP hydrolysis, although it does not prevent Drp1 from forming its classic ring structure around the mitochondria.

**Figure 10 cells-12-01897-f010:**
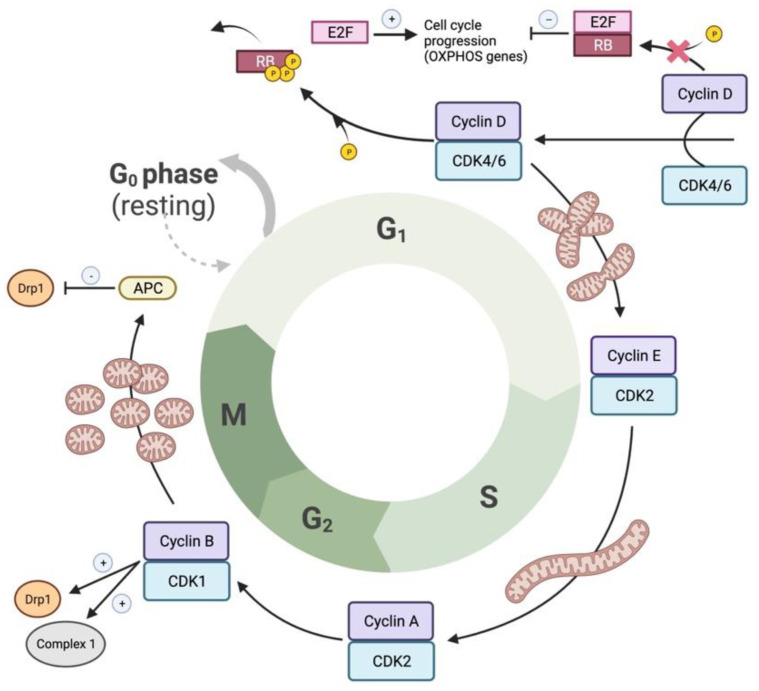
Coupling cell cycle regulation with mitochondrial function and dynamics. Activation of the cyclin D–CDK4/6 complex early in the G1 phase partially inactivates the pocket proteins, pRB, p107 and p130, releasing E2F transcription factor activity and, thus, allows the expression of genes necessary for the G1 to S transition. CDK2 is then activated, as a complex with cyclin E or A, during S and G2 phases, respectively. CDK1 then interacts with B type cyclins, which drives cells through mitosis. CDK1 promotes G2 to M transition by phosphorylating and activating complex I of the ETC and helps drive cells through mitosis by activating Drp1. The cycle will terminate upon activation of the APC complex and the subsequent degradation of type B cyclins and Drp1. Cyclin–CDK activity complex determines cell cycle progression, which is accompanied by changes in mitochondrial morphology. The mitochondrial network transforms from an interconnected network in G1 to a very fragmented network in mitosis, forming a hyperpolarized, giant single tubular network at the G1/S transition (in some cell types). Mitochondrial activity is also regulated by cell cycle regulators at the level of nuclear transcription. RB–E2F complexes regulate the transcription of mitochondrial genes. Positive regulators of cell cycle are in white. Negative regulators of cell cycle are in black.

**Figure 11 cells-12-01897-f011:**
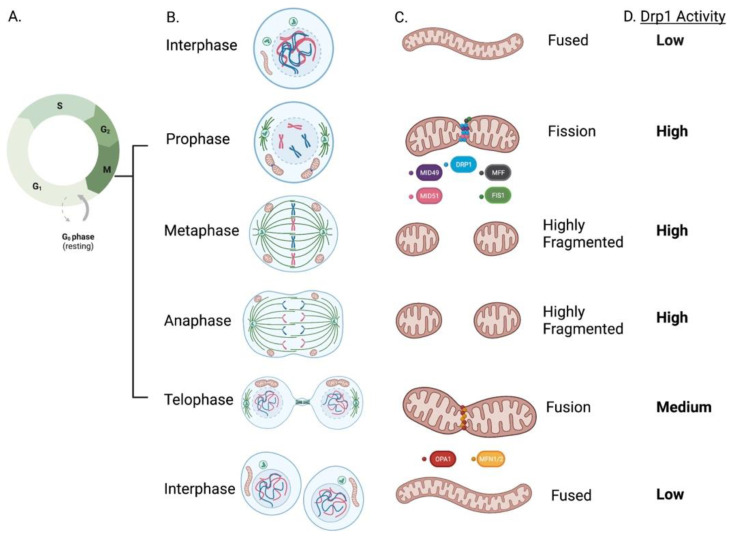
(**A**–**D**) Mitochondrial Dynamics During Mitotic (M phase) progression: Prophase is the first mitotic stage and is characterized by nuclear envelope breakdown, condensation of chromosomes and the beginning of mitotic spindle formation. At this stage, mitochondria prepare for fission by recruiting the Drp1 at the outer mitochondrial membrane through interactions with its receptors (MFF, Fis1, MiD49 and MiD51). During prophase, and peaking in metaphase, pro-fissive activities of Drp1 favor a highly fragmented network. As sister chromatids separate from each other during anaphase, mitochondria are still fully fragmented. During telophase, as the daughter cells separate, the APC complex has triggered degradation of fission mediators and the mitochondria begin to elongate again, through the tandem actions of pro-fusion proteins Mfn1/2 at the OMM and OPA1 at the IMM. Upon cytokinesis completion, mitochondria can be observed as a hyperfused, interconnected network distributed into the two new identical daughter cells.

**Figure 12 cells-12-01897-f012:**
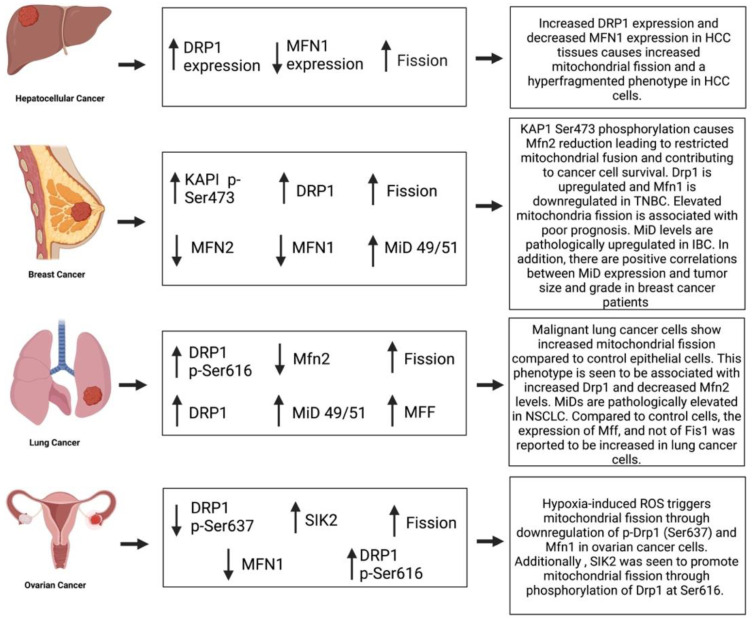
Summary of the pathological changes affecting mitochondrial dynamics in various cancers.

**Figure 13 cells-12-01897-f013:**
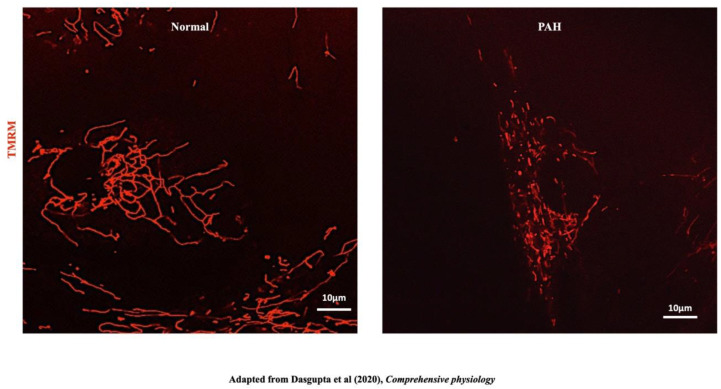
Mitochondria are fragmented in PAH. Representative images of mitochondrial networks of normal PASMC and PAH PASMC stained with the potentiometric dye TMRM (red) and imaged with confocal microscopy. Reproduced with permission from [87].

**Table 1 cells-12-01897-t001:** Mitochondrial Fission and Fusion Proteins.

Fusion Proteins	Mfn1
	Mfn2
	OPA1
Fission Proteins	Drp1
	Fis1
	MFF
	MiD49
	MiD51
	DNM2

## Data Availability

Not applicable.

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
