# Peer review of "The Role of Mitochondrial Dynamics and Mitotic Fission in Regulating the Cell Cycle in Cancer and Pulmonary Arterial Hypertension: Implications for Dynamin-Related Protein 1 and Mitofusin2 in Hyperproliferative Diseases"

_cells, 2023, doi:10.3390/cells12141897_

Round 1

Reviewer 1 Report

This is an exceptional comprehensive review on Drp1 and mitochondrial fusion regulation using the fields of some of the most active areas of investigation in cancer cells and pulmonary arterial hypertension to illustrate how it functions in important disease relevant processes. The only revision needed that i am providing as feedback for someone to correct is the omission of the D in downregulation in the first word in line 1016 of the second to the last paragraph of the manuscript text. Since I am not that good at careful proofing, there may be other minor things that need correction.

Author Response

General Comments.

This is an exceptional comprehensive review on Drp1 and mitochondrial fusion regulation using the fields of some of the most active areas of investigation in cancer cells and pulmonary arterial hypertension to illustrate how it functions in important disease relevant processes. The only revision needed that i am providing as feedback for someone to correct is the omission of the D in downregulation in the first word in line 1016 of the second to the last paragraph of the manuscript text. Since I am not that good at careful proofing, there may be other minor things that need correction.

General Response:

Thank you very much for your comments, the D has been added to downregulation as per your suggestion. The manuscript has been read again for minor corrections and updated as needed. Thank you for your support of our work and we look forward to finalizing this submission.

Reviewer 2 Report

The review manuscript of the authors of Colpman et al. aims to summarize the current state of knowledge in the field of mitochondrial fusion-fission dynamics and its relevance to selected types of cancer. Latest and future trends for effective treatment of cancer targeting the structures facilitating mitochondrial fusion and fission are involved as well – for example the Drpitor1/1a discovered by the authors recently. Indeed, focus on mitochondrial dynamics as effective treatment not only for the cancers, but in a broader context for a plethora of civilization diseases including diabetes, neurodegenerative and cardiovascular diseases and others merits further evaluation and development of new effective drugs is of critical importance and has broad application potential. In this case, the authors provide detailed and the latest knowledge about proteins (Mfn1/2, OPA1, Drp1, Fis1 and others), lipids (cardiolipin, phosphatidic acid) and processes (involvement of endoplasmic reticulum (ER), mitochondrial movement) orchestrating mitochondrial fusion and fission. Second part of the review aims to describe mitochondrial dynamics connected with lung, breast, ovarian and hepatocellular cancer as well as with pulmonary arterial hypertension (PAH), describing specific as well as common aspects of these pathologies. The authors cite relevant original literature. In general, the level of the review as whole represents high quality, robust and comprehensive review. However, there are several important points which should be addressed, see below.

Major comments (sorted by relevance):

11. The placement of Fig. 13 at the very end of the manuscript is confusing. In my opinion, this figure should be: A) omitted at all - this information is in principle available at the Figs. 1, 3 and 5; B) merged with some of the Figures mentioned above or C) to place this figure close to the text where it logically belongs.

22. Instead of Fig. 13 in its current form I would highly appreciate at the end of the review as a summary scheme for the process as a whole, i.e. mitochondrial fusion-fission in cancer and/or PAH.

33. Fig. 11 – the text in the rightmost boxes does not correspond to the type of cancer for the particular line (as example, for line with breast cancer in the right box ovarian cancer cells are discussed).

44.  Fig. 11 – protein KAP1 is stated (and listed in the Abbreviations as well), however, there is any information provided about KAP1 in the text. If crucial for the review, please provide relevant information about the KAP1.

55. Fig. 8 – I recommend to depict the cardiolipin (CL) and phosphatidic acid (PA) as lipids (i.e. hydrophobic tails and polar head) instead of simple circles. Moreover, the exact localization of CL and PA (outer and/or inner mitochondrial membrane) is not clear. In general, for this figure, I recommend to present the mitochondria partially only (inset, not the whole organelles) with higher magnification of the outer mitochondrial membrane – intermembrane space – inner mitochondrial membrane. In addition, it is not clear, why the ER is depicted at this figure, as any involvement of the ER is not demonstrated or discussed at this point. I would recommend removing the ER in this figure. Alternatively to implement its role for these processes and describe them shortly in the legend.

66. Authors should provide brief definition of MCT (page 10) its relevance in the context of the manuscript.

77. Similarly for SU5416 (page 26).

Minor issues:

Line 248 – does the term ’eliminated’ means suppressed or silenced? If yes, then I recommend to replace the word ’eliminated’ by more appropriate one

Line 538-540 – a reference would be appreciated for the arguments stated in this sentence

Line 849-850 – please specify at the end of the sentence which types of cancer cells were evaluated

Line 1054-1056 – reference should be included for this statement

Lines 893; 1000 – I recommend to use term ‘Warburg effect’ instead of ‘Warburg metabolism’

Formal issues:

Fig. 2 – the rectangle left side from the ’coil-coil region’ should be in green colour (now in blue)

Fig. 13 – if the authors decide to keep this figure (see the major comments), adjust the colour of the ER network, in part 1 it is presented in dark blue,  in the rest of the figure it appears in pale blue.

Figs. 2; 3; 4; 6; 7; 12 – the reference at the end of the Figure legend  - I recommend to insert the number of the reference as well, it is easier to find that particular reference in the Reference list

Lines 504-509; 887-897; 1012-1014; 1050-1052 – increase the font size as the rest of the text

Line 1062 – ref. Chakrabarti et al., 2018 should be numbered

Line 896 – ref. Gao et al., 2020 should be numbered

Lines 860; 1029 - the term ‘in vivo’ should be in Italics

Line 110, line 683 – correct ’are’ to ’is’

Line 160 – correct ’suggests’ to ’suggest’

Line 831 – correct ‘mitchondrial’ to ‘mitochondrial’

Line 1016 – letter ‘D’ is missing in the first word of the paragraph

In general, the level of English is appropriate.

Reviewer 3 Report

This is an excellent and comprehensive review article.  The main suggestion is a summary slide of the diseases studied and perhaps shortening the fusion/fission text as it is quite long.  

Author Response

General Comments:

This is an excellent and comprehensive review article. The main suggestion is a summary slide of the diseases studied and perhaps shortening the fusion/fission text as it is quite long.  

General Response:

Thank you very much for this suggestion. The fission/fusion section of the text has been shortened repeatedly and it is our thoughts that any effort to shorten it further would leave out key information in the text. A few words have been taken out following your review which were redundant, however most of the text has been kept in this section. If you strongly disagree with this decision, please let us know and we will make additional efforts to shorten this content.

As for the summary slide of diseases studied, this was thought to be accomplished with figure 12 which summarizes the pathological changes affecting mitochondrial dynamics in all Cancers mentioned. The only disease which is omitted from this section is PAH. PAH and cancer are different categories of pathology but if you think it would be better, we can include PAH in figure 12 also summarizing the various changes in mitochondrial dyanmics in a similar way to the cancer figure (Figure 12).

Round 2

Reviewer 2 Report

The authors have accepted and have successfully implemented most of the comments raised. However, there are still several minor issues which have to be addressed before the manuscript can be accepted in the MDPI Cells Journal.

Minor issues:

1.      Page 15, paragraph 2 (line 551). I recommend to change the beginning of the sentence ‘Cardiolipin is exclusively generated in the IMM by cardiolipin synthase ’ to ‘Cardiolipin is exclusively generated in the IMM by cardiolipin synthase from phosphatidic acid’.

2.      Page 15, paragraph 2 (line 552). Authors mention peroxisomes, are these organelles important in this context? If not, I would rather omit this topic from the text.

3.      MCT – authors have answered they have implemented a brief description of the MCT. Unfortunately, I cannot find where exactly to find this description. Moreover, the abbreviation MCT is missing in the Abbreviations Table.

4.      SU5416 – again, authors proclaim a description of this substance, unfortunately by checking the manuscript I am not able to find any new text added in this context. Please provide information (page, paragraph) where this information has been implemented.

5.      KAP1 is still mentioned in Fig. 12 (in the rightmost box for Breast Cancer). Please adjust accordingly.

6.      Page 17, paragraph 1 (lines 591-593) – after sentence ‘When the Drp1 domain that binds cardiolipin is mutated, “donut”-shaped circular mitochondria are generated, which may be important  in stress-induced fission.’ a reference should be added.

7.      Page 24, paragraph 1 (lines 918-920) – at the end of the sentence ‘Our group has reported mitochondrial hyperpolarization as a hallmark of both PAH and cancer cells’ the information which cancer cells exactly this phenomenon has been observed in.

8.      Page 9, Fig. 5 Legend – a reference for the sentence ‘Mitochondrial DNA (mtDNA) replication marks the site for the recruitment of the endoplasmic reticulum (ER) to contact the mitochondria at a site of impending midzone fission.’ should be added.

Formal issues:

Page 14, sentence ‘Communication between mitochondrial fission and fusion mediators’ – is this a sub-title? If yes, then please mark it in correct form. If not, then merge with the paragraph.

Please correct all through the manuscript the term ‘in vivo’ in Italics font (lines 445, 485, 512, 930, 1111)

Page 8, paragraph 2 (line 287) and Page 17, paragraph 3 (line 618) – correct ‘figure 5’ to ‘Figure 5’

In all Figures the first sentence of the Figure legend should be in Bold font. Please correct.

Page 16, Figure legend, please correct ‘and Cardiolipin Phosphatidic acid’ to ‘and Cardiolipin. Phosphatidic acid’

Page 15, paragraph 2 (line 551), please correct ‘Cardiolipin, is’ to ‘Cardiolipin is’

Page 4, Table 1 Legend. Please correct ’Table 1. 0: Mitochondrial Fission and Fusion Proteins’ to ’Table 1. Mitochondrial Fission and Fusion Proteins.’
